# LLM-QAT: Data-Free Quantization Aware Training for Large Language Models

## Abstract

Several post-training quantization methods have been applied to large language models (LLMs), and have been shown to perform well down to 8-bits. We find that these methods break down at lower bit precision, and investigate quantization aware training for LLMs (LLM-QAT) to push quantization levels even further. We propose a data-free distillation method that leverages generations produced by the pre-trained model, which better preserves the original output distribution and allows quantizing any generative model independent of its training data, similar to post-training quantization methods. In addition to quantizing weights and activations, we also quantize the KV cache, which is critical for increasing throughput and support long sequence dependencies at current model sizes. We experiment with LLaMA models of sizes 7B, 13B, and 30B, at quantization levels down to 4-bits. We observe large improvements over training-free methods, especially in the low-bit settings.

## 1 Introduction

Following GPT-3 (Brown et al., 2020), several families of large language models (LLMs) such as OPT (Zhang et al., 2022), PALM (Chowdhery et al., 2022), BLOOM (Scao et al., 2022), Chinchilla (Hoffmann et al., 2022) and LLaMA (Touvron et al., 2023) have established that increasing model size leads to improved model capabilities. As a result, language models with tens of billions or even hundreds of billions of parameters have become the norm in today's AI landscape. Despite the growing excitement around LLMs, serving such models to the benefit of billions of users faces significant hurdles due to their large computational cost and environmental footprint.

Fortunately, there has been an increasing effort to accurately quantize LLMs, with multiple recent works (Xiao et al., 2022; Yao et al., 2022) focusing on 8-bit post-training quantization of weights and activations and achieving little to no loss of accuracy. as well as quantizing the weight and KV cache and using GPU/CPU offloading to achieve high-throughput LLM inference (Sheng et al., 2023a) . However, SoTA post-training quantization methods dramatically degrade in quality when quantizing weights, activations and KV cache together to below 8-bit. For lower quantization bit-widths, we find it necessary to resort to quantization-aware training (QAT).

To our knowledge, QAT for LLMs has not been investigated before. This is understandable for two reasons. First, LLM training is technically difficult and resource intensive. Second, QAT needs training data, which for LLMs is difficult to obtain. The sheer scale and diversity of pre-training data is itself an obstacle. Pre-processing might be prohibitive, or worse, some data might simply not be available due to legal restrictions. It is also increasingly common to train LLMs in multiple stages, involving instruction tuning and reinforcement learning (Ouyang et al., 2022), which would be very difficult to replicate during QAT. In this work, we side-step this issue by using generated data from the LLM itself for knowledge distillation. This simple workaround, which we refer to as *data-free* knowledge-distillation is applicable to any generative model independent of whether or not the original training data is available. We show that this method is better able to preserve the original model's output distribution, even compared to training on large subsets of the original training set. Moreover, we can successfully distill quantized models using only a small set (100k) of sampled data, thus keeping computational costs reasonable. All of our experiments are conducted using a single 8-gpu training node.

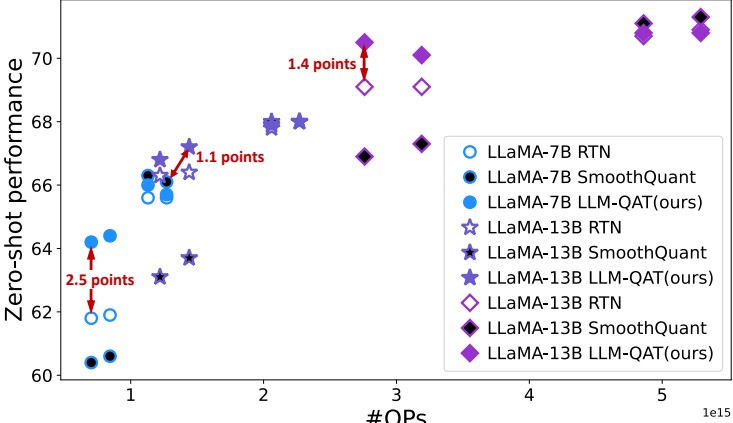

Figure 1: Employing LLM-QAT, accuracy improves by 2.5 and 1.4 points for LLaMA-7B and LLaMA-30B models over SoTA PTQ under W4A8KV8 settings. Additionally, W4A8KV16 LLaMA-13B outperforms W8A8KV16 LLaMA-7B by 1.1 points with similar OPs. These improvements are substantial, especially when considering that the LLaMA-13B model surpasses the performance of the 7B model by a mere 1.8 points. The quantization settings are W4A8KV8 / W4A8KV16 / W8A8KV8 / W8A8KV16 from left to right. See Table 1 for details.

As a result, we are able to distill the 7B, 13B and 30B LLaMA models with activations quantized down to 8 bits, weights and KV cache down to 4-bits. In this regard, our approach exhibits significant enhancements in quality compared to post-training quantization. Notably, larger models employing QAT outperform smaller models utilizing floating-point 16-bit representations, despite having similar model sizes, as illustrated in Figure 1. Furthermore, we have successfully quantized activations to 6-bit precision, surpassing what was possible with existing methods. For a comprehensive analysis of our experimental results and detailed ablations, please refer to Section 3.

In summary, we present the first application of QAT to LLMs, resulting in the first accurate 4-bit quantized LLMs. We also demonstrate quantizing the KV cache simultaneously with weights and activations, which is critical to alleviate throughput bottlenecks for long sequence generation. All of this is achieved by a novel *data-free* distillation method, which makes QAT practical for large pre-trained generative models.

## 2 METHOD

Quantizing large language models (LLMs) using quantization-aware training (QAT) is a nontrivial task with challenges in two key aspects. First, LLMs are pre-trained to excel in zero-shot generalization, and it is crucial to preserve this capability after quantization. Therefore, selecting an appropriate fine-tuning dataset is important. If the QAT data is too narrow in domain or significantly different than the original pre-training distribution, this is likely to hurt model performance. On the other hand, it is difficult to replicate the original training setup exactly, due to the scale and complexity of LLM training. In Section 2.1, we introduce *data-free* quantization-aware training (QAT) which produces QAT data using next token data generation. This method demonstrates superior performance compared to using subsets of the original pre-training data. Second, LLMs exhibit unique weight and activation distributions characterized by a significant presence of outliers, which distinguishes them from smaller models. Consequently, the state-of-the-art quantization clipping methods for small models do not work out of the box for LLMs. In Section 2.2, we identify suitable quantizers for LLMs.

### 2.1 DATA-FREE DISTILLATION

In order to closely synthesize the distribution of the pre-training data with a limited amount of fine-tuning data, we proposed next token data generation from the original pre-trained model. As shown in Figure 2 (a), we randomize the first token $<start>$ from vocabulary and let the pre-trained

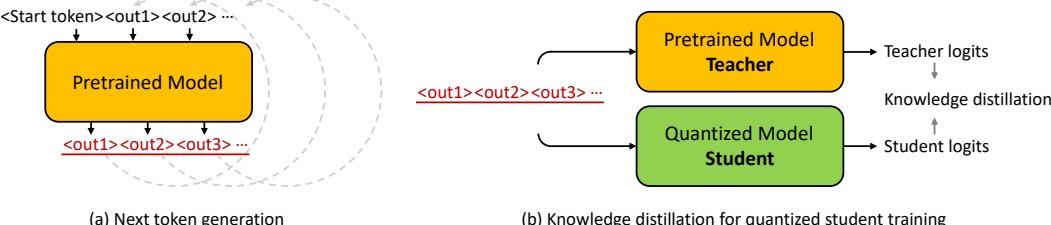

Figure 2: Overview of `LLM-QAT`. We generate data from the pretrained model with next token generation, which is sampled from top-k candidates. Then we use the generated data as input and the teacher model prediction as label to guide quantized model finetuning.

model to generate the next token $<out1>$, then the generated token is appended to the start token for generating new output $<out2>$. We repeat this iterative procedure until we reach either the end of sentence token or the maximum generation length.

We test three different sampling strategies in the next token generation. The most straightforward way is to pick the top-1 candidate as the next token. However, the generated sentence lacks of diversity and will cyclically repeat several tokens. To address this issue, we instead stochastically sample the next token from the distribution using the SoftMax output of the pre-trained model as the probability. This sampling strategy yields more diverse sentences and greatly enhances the accuracy of the fine-tuned student model. Furthermore, we discover that the initial few tokens play a crucial role in determining the prediction trend. Therefore, it is important for them to have higher confidence. In our generative process, we employ a hybrid sampling strategy that deterministically selects the top-1 predictions for the first 3~5 tokens and stochastically samples the remaining tokens. A detailed ablation study comparing different generated data and real data is presented in Section3.3.1.

## 2.2 QUANTIZATION-AWARE TRAINING

### 2.2.1 PRELIMINARIES

In this work, we study linear quantization *i.e.*, uniform quantization. Linear quantization can be categorized into two categories based on whether the real values are clipped or not: MinMax quantization, which preserves all value ranges, and clipping-based quantization.

In MinMax quantization, the quantization process can be formulated as:

$$\mathbf{X_Q}^i = \alpha \hat{\mathbf{X}}_{\mathbf{Q}}^i = \alpha \lfloor \frac{\mathbf{X_R}^i - \beta}{\alpha} \rceil + \beta. \tag{1}$$

Here $\mathbf{X_Q}$ and $\mathbf{X_R}$ denote the quantized and full-precision variables, respectively. $i$ refers to the $i$-th element in the tensor. $\alpha$ is the scaling factor and $\beta$ is the zero-point value. For symmetric quantization, $\alpha = \frac{\max(|\mathbf{X_R}|)}{2^{N-1}-1}$, $\beta = 0$. And for asymmetric quantization, $\alpha = \frac{\max(\mathbf{X_R})-\min(\mathbf{X_R})}{2^N-1}$, $\beta = \min(\mathbf{X_R})$.

Compared to the MinMax Quantization, clipping the outliers can help improve the precision and allocate more bits to the intermediate values. Thus, many recent work (Shen et al., 2020a; Zhang et al., 2020) adopts clipping-based quantization for transformer-based language models. The quantization can be formulated as:

$$\mathbf{X_Q}^i = \alpha \hat{\mathbf{X}}_{\mathbf{Q}}^i = \alpha \lfloor \text{Clip}(\frac{\mathbf{X_R}^i - \beta}{\alpha}, 0, 1) \rceil + \beta. \tag{2}$$

where the scale $\alpha$ and zero-point value $\beta$ can be calculated statistically or learned through gradients.

### 2.2.2 QUANTIZATION FOR LARGE LANGUAGE MODELS

**Quantization function** We illustrate our quantized transformer model in Figure 3. In line with the findings in (Dettmers et al., 2022; Xiao et al., 2022), we have also observed a significant presence of outliers in both the weights and activations of large language models (LLMs). These outliers have a notable impact on the quantization process, as they contribute to an increase in the quantization step size while diminishing the precision of intermediate values. Nevertheless, clipping these outliers

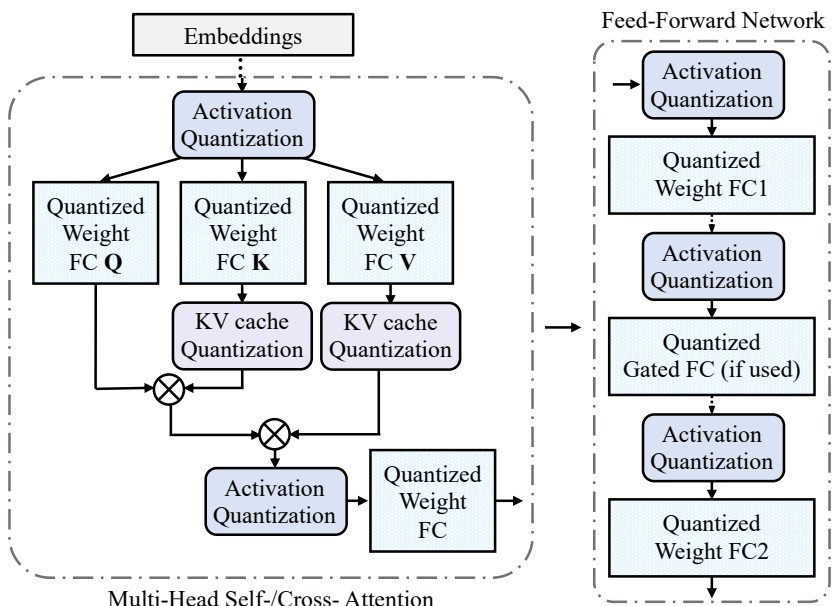

Figure 3: Overview of the quantized transformer in `LLM-QAT`. We quantize all the weights and input activations in fully-connected linear layers. The KV cache is also quantized if specified.

during quantization proves detrimental to LLM performance. During the initial stages of training, any clipping-based method will lead to exceptionally high perplexity scores (*i.e.,* $> 10000$), causing a substantial loss of information that proves to be difficult to recover through fine-tuning. Therefore, we choose to retain these outliers instead. Moreover, we find that in the model with the gated linear unit (GLU), the activations are weights are mostly symmetrically distributed. Based on our analysis and empirical observations, we choose symmetric MinMax quantization for both weights and activations:

$$\mathbf{X}_{\mathbf{Q}}^i = \alpha \lfloor \frac{\mathbf{X}_{\mathbf{R}}^i}{\alpha} \rceil, \quad \alpha = \frac{\max(|\mathbf{X}_{\mathbf{R}}|)}{2^{N-1} - 1} \tag{3}$$

Here $\mathbf{X}_{\mathbf{Q}}$ denotes the quantized weights or activations and $\mathbf{X}_{\mathbf{R}}$ denotes the real-valued weights or activations. To ensure efficient quantization, we adopt the per-token activation quantization and per-channel weight quantization. For a comprehensive evaluation of the different quantizer choices, we provide the ablation study in Section 3.3.2.

**Quantization-aware training for key-value cache** In addition to weight and activation quantization, the key-value cache (KV cache) in large language models (LLMs) also consumes a non-negligible amount of memory. However, only a few previous works have addressed the KV cache quantization in LLMs, with the methods primarily limited to post-training quantization(Sheng et al., 2023b). In our study, we demonstrate that a similar quantization-aware training approach used for activation quantization can be employed to quantize the KV cache. We adopt per-token quantization in Eq. 3, given that the key and value are generated token by token. During the generation process, the current key and value are quantized, and their corresponding scaling factor is stored. During the training process for QAT, we apply quantization to the entire activation tensors of both the keys and values, as shown in Figure 3. By integrating the quantization function into the gradient calculation, we ensure effective training using quantized key-value pairs.

**Knowledge distillation** We use cross-entropy based logits distillation for training the quantized student network from the full-precision pre-trained teacher network:

$$\mathcal{L}_{CE} = -\frac{1}{n} \sum_c \sum_{i=1}^n p_c^{\mathcal{T}}(X_i) \log(p_c^{\mathcal{S}}(X_i)), \tag{4}$$

Here $i$ denotes the $i^{th}$ sample in the current batch with $n$ total sentences. $c$ denotes the number of classes and in our case, it equals the size of the vocabulary. $\mathcal{T}$ and $\mathcal{S}$ are the teacher network and student network, respectively.

As discussed in Section 2.1, in the data generation process, it is important to sample the next token from distribution rather than always selecting the top-1 candidate. By doing so, the next token does not necessarily represent the optimal label for training the student model, as the sampling introduces inherent noise. Consequently, we propose to utilize the predictions from the pre-trained model as soft labels, which provides more informative targets for guiding the training of the student model. We present a comprehensive ablation study in Section 3.3.2 to delve into the specifics of this approach.

# 3 EXPERIMENTS

We assess the effectiveness of our approach by conducting experiments on LLaMA-7B/13B/30B models and presenting results on various tasks. Specifically, we report the zero-shot performance on Common Sense Reasoning tasks such as BoolQ (Clark et al., 2019), PIQA (Bisk et al., 2020), SIQA (Sap et al., 2019), HellaSwag (Zellers et al., 2019), WinoGrande (Sakaguchi et al., 2021), ARC (Clark et al., 2018), and OBQA (Mihaylov et al., 2018). We also assess the few-shot performance on TriviaQA (Joshi et al., 2017) and MMLU (Hendrycks et al., 2020) datasets, along with perplexity scores on WikiText2 (Merity et al., 2016) and C4 (Raffel et al., 2020) datasets.

## 3.1 EXPERIMENTAL SETTINGS

In our quantized network training process, we initialize the model with a pre-trained model and employ it as the teacher for knowledge distillation. To optimize the model, we utilize the AdamW (Loshchilov & Hutter, 2017) optimizer with zero weight decay. Each GPU is assigned a batch size of 1, and the learning rate is set to 2e-5, following a cosine learning-rate decay strategy. For data generation, we utilize the LLaMA-7B model, and the maximum length of generated sequences is set to 2048. We calculate number of OPs by $\text{OPs} = \text{MACs} \times \text{W}_{\text{bits}} \times \text{A}_{\text{bits}}$ with the sequence length equals to 2048.

## 3.2 MAIN RESULTS

We consider three post-training quantization (PTQ) methods, round-to-nearest (RTN), GPT-Q (Frantar et al., 2022) and SmoothQuant (Xiao et al., 2022) as baselines. We compare to them in several different settings, where the weights, activations and KV cache values are quantized to different levels (denoted as W-A-KV). Different PTQ methods perform well in different settings, and we compare our method to the best PTQ result in each setting.

Table 1, table 2 and table 7 (in Appendix) give the comparisons of the proposed QAT methods with SOTA PTQ methods for LLMs on Zero-shot tasks on Common Sense Reasoning tasks, perplexity evaluation on Wiki2 and C4 and few shot exact match on the MMLU and TriviaQA benchmarks respectively. The perplexity evaluations verify whether the quantize models are able to preserve the output distribution of the model on a diverse sample of its training domains. The zero-shot and few-shot evaluations measure if the model's capabilities on downstream tasks are retained.

The trends in each table are similar. All methods tend to do well in the 8-bit setting across all model sizes. This holds even when the KV cache is also quantized to 8-bits, together with weights and activations. However, when either of these three values are quantized to less than 8-bits, PTQ methods result in accuracy loss, whereas LLM-QAT holds up much better. For example in the 8-8-4 setting, 30B LLM-QAT achieves an average zero-shot accuracy of 69.7, compared to 50.7 with SmoothQuant (Table 1, rows 68-69). The difference is smaller in the 4-8-8 setting, however LLM-QAT still outperforms the best PTQ method (RTN in this case) by 1.4 points (rows 55, 57). In the 4-8-4 setting, where both weights and the KV cache are quantized to 4 bits, all PTQ methods produce poor results, whereas LLM-QAT achieves 69.9, only trailing the full precision model by 1.5 points on average. LLM-QAT also works reasonably well for 6-bit activation quantization. While this setting might not be currently practical due to lack of hardware support, it's a promising data point for sub-8-bit computation for LLMs.

One important question for practitioners is whether to use a small model at full precision, or a larger quantized model of similar inference cost. While the exact trade-offs can vary based on several factors, we can make several recommendations based on our results. First, 8-bit quantization should be preferred over smaller full precision models, and PTQ methods are sufficient for this case. An 8-8-8 30B quantized model outperforms a 13B model of similar size, and should have lower latency

Table 1: Zero-shot performance on Common Sense Reasoning tasks.

| | Method | #Bits | #OPs (×10^15) | Size (GB) | BoolQ (↑) | PIQA (↑) | SIQA (↑) | HellaSwag (↑) | WinoGrande (↑) | ARC-e (↑) | ARC-c (↑) | OBQA (↑) | Avg. (↑) |
|---|---|---|---|---|---|---|---|---|---|---|---|---|---|
| 1 | LLaMA-7B | 16-16-16 | 3.81 | 12.6 | 76.8 | 79.3 | 48.6 | 76.1 | 70.0 | 73.0 | 48.0 | 57.6 | 66.2 |
| 2 | RTN | 4-8-4 | 0.63 | 3.5 | 51.9 | 56.3 | 40.5 | 35.7 | 49.9 | 39.3 | 25.3 | 30.8 | 41.2 |
| 3 | SmoothQuant | 4-8-4 | 0.63 | 3.5 | 54.7 | 55.4 | 41.1 | 38.9 | 51.5 | 43.9 | 27.7 | 32.0 | 43.2 |
| 4 | LLM-QAT | 4-8-4 | 0.63 | 3.5 | 69.5 | 75.4 | 46.6 | 69.2 | 64.6 | 66.0 | 43.8 | 50.6 | **60.7** |
| 5 | RTN | 4-8-8 | 0.70 | 3.5 | 67.8 | 76.6 | 47.2 | 71.4 | 67.2 | 67.4 | 45.6 | 51.2 | 61.8 |
| 6 | SmoothQuant | 4-8-8 | 0.70 | 3.5 | 71.0 | 76.0 | 45.4 | 67.8 | 66.0 | 67.4 | 42.8 | 47.0 | 60.4 |
| 7 | LLM-QAT | 4-8-8 | 0.70 | 3.5 | 74.6 | 77.5 | 48.3 | 73.5 | 67.7 | 70.2 | 45.6 | 56.2 | **64.2** |
| 8 | RTN | 4-6-16 | 0.74 | 3.5 | 62.4 | 74.5 | 46.8 | 67.9 | 64.5 | 64.6 | 41.5 | 49.0 | 58.9 |
| 9 | SmoothQuant | 4-6-16 | 0.74 | 3.5 | 68.8 | 73.9 | 44.5 | 65.7 | 65.3 | 66.0 | 43.6 | 48.0 | 59.5 |
| 10 | LLM-QAT | 4-6-16 | 0.74 | 3.5 | 72.9 | 76.8 | 47.9 | 72.4 | 68.3 | 68.8 | 44.2 | 53.2 | **63.1** |
| 11 | RTN | 4-8-16 | 0.84 | 3.5 | 67.6 | 77.4 | 47.1 | 71.6 | 66.9 | 67.1 | 45.8 | 52.0 | 61.9 |
| 12 | SmoothQuant | 4-8-16 | 0.84 | 3.5 | 70.2 | 76.4 | 44.8 | 68.1 | 66.0 | 67.3 | 42.9 | 49.0 | 60.6 |
| 13 | LLM-QAT | 4-8-16 | 0.84 | 3.5 | 74.8 | 77.8 | 48.6 | 73.6 | 69.0 | 69.7 | 45.8 | 55.8 | **64.4** |
| 14 | RTN | 4-16-16 | 1.27 | 3.5 | 71.2 | 77.3 | 47.6 | 72.7 | 66.9 | 68.8 | 46.4 | 52.8 | 63.0 |
| 15 | GPTQ | 4-16-16 | 1.27 | 3.5 | 67.7 | 76.0 | 46.8 | 69.4 | 66.7 | 66.9 | 43.0 | 50.6 | 60.9 |
| 16 | LLM-QAT | 4-16-16 | 1.27 | 3.5 | 75.5 | 78.3 | 48.4 | 74.0 | 69.0 | 70.0 | 45.0 | 55.4 | **64.4** |
| 17 | RTN | 8-8-4 | 1.06 | 6.5 | 54.7 | 59.4 | 43.1 | 45.6 | 57.4 | 51.2 | 29.6 | 37.8 | 47.4 |
| 18 | SmoothQuant | 8-8-4 | 1.06 | 6.5 | 60.7 | 67.5 | 44.9 | 58.3 | 58.6 | 57.5 | 36.9 | 43.6 | 53.5 |
| 19 | LLM-QAT | 8-8-4 | 1.06 | 6.5 | 71.1 | 75.6 | 47.3 | 71.8 | 66.3 | 67.1 | 43.6 | 50.0 | **61.6** |
| 20 | RTN | 8-8-8 | 1.13 | 6.5 | 76.4 | 79.5 | 48.7 | 75.5 | 69.5 | 72.3 | 46.6 | 56.0 | 65.6 |
| 21 | SmoothQuant | 8-8-8 | 1.13 | 6.5 | 76.1 | 79.6 | 48.7 | 76.2 | 70.1 | 73.7 | 48.7 | 57.0 | **66.3** |
| 22 | LLM-QAT | 8-8-8 | 1.13 | 6.5 | 76.0 | 79.6 | 48.5 | 75.7 | 69.4 | 73.1 | 48.2 | 57.4 | 66.0 |
| 23 | RTN | 8-8-16 | 1.27 | 6.5 | 76.4 | 79.1 | 48.3 | 75.7 | 70.5 | 72.8 | 46.5 | 55.6 | 65.6 |
| 24 | SmoothQuant | 8-8-16 | 1.27 | 6.5 | 76.2 | 79.5 | 48.6 | 76.1 | 70.5 | 73.2 | 47.7 | 57.2 | **66.1** |
| 25 | LLM-QAT | 8-8-16 | 1.27 | 6.5 | 76.3 | 79.4 | 48.7 | 75.6 | 69.7 | 72.3 | 47.6 | 56.2 | 65.7 |
| 26 | LLaMA-13B | 16-16-16 | 7.26 | 24.2 | 78.1 | 80.0 | 50.5 | 79.2 | 73.6 | 74.5 | 52.6 | 55.0 | 68.0 |
| 27 | RTN | 4-8-4 | 1.11 | 6.5 | 54.0 | 59.2 | 41.9 | 41.6 | 55.9 | 45.0 | 27.0 | 33.2 | 44.7 |
| 28 | SmoothQuant | 4-8-4 | 1.11 | 6.5 | 63.0 | 65.3 | 42.2 | 50.6 | 54.1 | 49.6 | 30.3 | 34.2 | 48.7 |
| 29 | LLM-QAT | 4-8-4 | 1.11 | 6.5 | 72.0 | 76.8 | 49.2 | 73.6 | 66.5 | 69.3 | 46.9 | 52.8 | **63.4** |
| 30 | RTN | 4-8-8 | 1.22 | 6.5 | 76.2 | 78.8 | 49.3 | 76.2 | 69.9 | 72.2 | 50.7 | 56.8 | 66.3 |
| 31 | SmoothQuant | 4-8-8 | 1.22 | 6.5 | 72.5 | 77.1 | 47.2 | 74.3 | 69.5 | 67.4 | 43.3 | 53.4 | 63.1 |
| 32 | LLM-QAT | 4-8-8 | 1.22 | 6.5 | 77.5 | 79.1 | 48.6 | 77.5 | 70.6 | 73.0 | 51.9 | 56.2 | **66.8** |
| 33 | RTN | 4-6-16 | 1.24 | 6.5 | 71.8 | 74.1 | 47.7 | 70.2 | 65.1 | 69.3 | 44.1 | 45.6 | 61.0 |
| 34 | SmoothQuant | 4-6-16 | 1.24 | 6.5 | 70.6 | 76.3 | 47.9 | 73.1 | 68.5 | 65.9 | 43.3 | 52.6 | 62.3 |
| 35 | LLM-QAT | 4-6-16 | 1.24 | 6.5 | 75.4 | 79.3 | 48.4 | 76.5 | 69.2 | 73.1 | 48.6 | 53.4 | **65.5** |
| 36 | RTN | 4-8-16 | 1.44 | 6.5 | 76.8 | 79.1 | 49.1 | 76.3 | 70.5 | 72.6 | 49.8 | 56.6 | 66.4 |
| 37 | SmoothQuant | 4-8-16 | 1.44 | 6.5 | 72.5 | 77.9 | 47.6 | 74.2 | 69.7 | 68.2 | 45.0 | 54.2 | 63.7 |
| 38 | LLM-QAT | 4-8-16 | 1.44 | 6.5 | 77.7 | 79.3 | 48.4 | 77.5 | 70.6 | 73.5 | 53.0 | 57.4 | **67.2** |
| 39 | RTN | 4-16-16 | 2.27 | 6.5 | 77.4 | 79.1 | 49.2 | 76.8 | 70.5 | 72.6 | 51.2 | 54.2 | 66.4 |
| 40 | GPTQ | 4-16-16 | 2.27 | 6.5 | 78.0 | 79.8 | 49.2 | 77.7 | 72.6 | 73.2 | 50.6 | 55.4 | **67.1** |
| 41 | LLM-QAT | 4-16-16 | 2.27 | 6.5 | 77.7 | 79.4 | 49.1 | 77.7 | 71.5 | 72.8 | 52.0 | 53.8 | 66.7 |
| 42 | RTN | 8-8-4 | 1.95 | 12.4 | 65.8 | 66.2 | 43.9 | 56.7 | 57.3 | 58.2 | 34.5 | 42.6 | 53.2 |
| 43 | SmoothQuant | 8-8-4 | 1.95 | 12.4 | 66.6 | 71.7 | 44.8 | 61.1 | 61.0 | 63.4 | 38.3 | 43.6 | 56.3 |
| 44 | LLM-QAT | 8-8-4 | 1.95 | 12.4 | 74.9 | 78.3 | 48.0 | 75.7 | 68.9 | 71.9 | 51.1 | 54.2 | **65.4** |
| 45 | RTN | 8-8-8 | 2.06 | 12.4 | 77.8 | 80.0 | 50.8 | 78.9 | 72.6 | 74.5 | 52.1 | 55.6 | 67.8 |
| 46 | SmoothQuant | 8-8-8 | 2.06 | 12.4 | 78.3 | 80.3 | 50.8 | 79.2 | 73.2 | 74.8 | 52.4 | 55.4 | **68.0** |
| 47 | LLM-QAT | 8-8-8 | 2.06 | 12.4 | 78.7 | 80.4 | 50.1 | 79.1 | 73.2 | 74.8 | 51.7 | 55.4 | 67.9 |
| 48 | RTN | 8-8-16 | 2.27 | 12.4 | 77.8 | 80.1 | 50.6 | 78.9 | 73.5 | 74.9 | 51.9 | 56.4 | **68.0** |
| 49 | SmoothQuant | 8-8-16 | 2.27 | 12.4 | 78.7 | 80.0 | 50.6 | 79.1 | 73.4 | 74.8 | 51.4 | 56.0 | **68.0** |
| 50 | LLM-QAT | 8-8-16 | 2.27 | 12.4 | 78.5 | 80.4 | 50.6 | 79.0 | 72.8 | 74.2 | 52.9 | 55.8 | **68.0** |
| 51 | LLaMA-30B | 16-16-16 | 17.9 | 60.6 | 83.2 | 82.1 | 50.4 | 82.9 | 75.6 | 80 | 58 | 59.3 | 71.4 |
| 52 | RTN | 4-8-4 | 2.54 | 15.7 | 56.9 | 56.2 | 40.2 | 39.6 | 50.0 | 40.6 | 26.4 | 29.8 | 42.5 |
| 53 | SmoothQuant | 4-8-4 | 2.54 | 15.7 | 56.6 | 55.0 | 39.9 | 33.8 | 49.9 | 38.8 | 24.5 | 27.2 | 40.7 |
| 54 | LLM-QAT | 4-8-4 | 2.54 | 15.7 | 80.5 | 80.3 | 49.7 | 80.2 | 75.2 | 78.2 | 56.0 | 59.2 | **69.9** |
| 55 | RTN | 4-8-8 | 2.76 | 15.7 | 78.8 | 79.9 | 49.0 | 80.2 | 75.2 | 78.4 | 54.4 | 57.2 | 69.1 |
| 56 | SmoothQuant | 4-8-8 | 2.76 | 15.7 | 74.9 | 79.5 | 47.1 | 76.9 | 70.6 | 76.5 | 54.5 | 55.0 | 66.9 |
| 57 | LLM-QAT | 4-8-8 | 2.76 | 15.7 | 81.3 | 80.9 | 50.4 | 81.3 | 76.3 | 80.3 | 56.5 | 57.0 | **70.5** |
| 58 | RTN | 4-6-16 | 2.66 | 15.7 | 64.5 | 57.0 | 42.1 | 48.9 | 55.4 | 39.3 | 27.0 | 32.2 | 45.8 |
| 59 | SmoothQuant | 4-6-16 | 2.66 | 15.7 | 75.0 | 77.6 | 46.6 | 73.8 | 69.1 | 74.5 | 52.9 | 50.6 | 65.0 |
| 60 | LLM-QAT | 4-6-16 | 2.66 | 15.7 | 78.8 | 80.3 | 50.3 | 79.9 | 75.1 | 77.0 | 54.4 | 59.0 | **69.4** |
| 61 | RTN | 4-8-16 | 3.19 | 15.7 | 79.1 | 79.6 | 49.5 | 80.4 | 74.9 | 78.3 | 53.7 | 57.2 | 69.1 |
| 62 | SmoothQuant | 4-8-16 | 3.19 | 15.7 | 76.0 | 79.8 | 48.2 | 77.0 | 71.6 | 76.4 | 55.6 | 54.2 | 67.3 |
| 63 | LLM-QAT | 4-8-16 | 3.19 | 15.7 | 80.6 | 80.8 | 50.1 | 81.2 | 75.8 | 79.7 | 56.3 | 56.3 | **70.1** |
| 64 | RTN | 4-16-16 | 5.29 | 15.7 | 80.8 | 80.1 | 49.8 | 81.6 | 75.8 | 79.3 | 55.8 | 57.2 | 70.1 |
| 65 | GPTQ | 4-16-16 | 5.29 | 15.7 | 81.0 | 81.6 | 49.7 | 82.2 | 74.3 | 79.6 | 56.1 | 58.2 | **70.3** |
| 66 | LLM-QAT | 4-16-16 | 5.29 | 15.7 | 81.8 | 81.0 | 49.7 | 81.8 | 75.1 | 79.4 | 56.8 | 54.9 | 70.1 |
| 67 | RTN | 8-8-4 | 4.65 | 30.7 | 59.8 | 64.5 | 42.7 | 51.8 | 55.0 | 52.2 | 33.2 | 38.0 | 49.6 |
| 68 | SmoothQuant | 8-8-4 | 4.65 | 30.7 | 58.9 | 64.4 | 43.5 | 54.8 | 55.2 | 55.3 | 33.6 | 40.2 | 50.7 |
| 69 | LLM-QAT | 8-8-4 | 4.65 | 30.7 | 81.2 | 81.6 | 50.1 | 81.1 | 73.6 | 78.5 | 55.7 | 55.7 | **69.7** |
| 70 | RTN | 8-8-8 | 4.86 | 30.7 | 82.2 | 81.2 | 49.4 | 81.9 | 75.6 | 79.6 | 57.4 | 58.2 | 70.7 |
| 71 | SmoothQuant | 8-8-8 | 4.86 | 30.7 | 82.5 | 82.3 | 50.2 | 82.8 | 75.9 | 80.3 | 56.9 | 57.8 | **71.1** |
| 72 | LLM-QAT | 8-8-8 | 4.86 | 30.7 | 82.2 | 81.3 | 51.0 | 82.3 | 75.0 | 80.2 | 57.0 | 57.2 | 70.8 |
| 73 | RTN | 8-8-16 | 5.29 | 30.7 | 82.3 | 81.6 | 50.2 | 81.7 | 75.9 | 79.7 | 56.7 | 59.0 | 70.9 |
| 74 | SmoothQuant | 8-8-16 | 5.29 | 30.7 | 82.8 | 81.9 | 50.3 | 82.7 | 76.3 | 80.2 | 57.7 | 58.4 | **71.3** |
| 75 | LLM-QAT | 8-8-16 | 5.29 | 30.7 | 82.4 | 81.4 | 50.3 | 82.5 | 76.0 | 80.0 | 57.2 | 56.8 | 70.8 |

Table 2: Perplexity evaluation results on WikiText Merity et al. (2016) and C4 Raffel et al. (2020)

| | #Bits | Method | Perplexity | | Method | Perplexity | | Method | Perplexity | |
|---|---|---|---|---|---|---|---|---|---|---|
| | | | C4 (↓) | Wiki2 (↓) | | C4 (↓) | Wiki2 (↓) | | C4 (↓) | Wiki2 (↓) |
| 1 | 16-16-16 | LLaMA-7B | 7.2 | 10.4 | LLaMA-13B | 6.7 | 9.7 | LLaMA-30B | 6.0 | 7.0 |
| 2 | 4-8-4 | RTN | 55.1 | 151.4 | RTN | 25.0 | 103.6 | RTN | 8.2 | 8.9 |
| 3 | 4-8-4 | SmoothQuant | 81.1 | 163.6 | SmoothQuant | 26.0 | 60.1 | SmoothQuant | 10.6 | 12.0 |
| 4 | 4-8-4 | LLM-QAT | **8.6** | **11.6** | LLM-QAT | **7.6** | **10.2** | LLM-QAT | **7.3** | **7.7** |
| 5 | 4-8-8 | RTN | 8.4 | 13.9 | RTN | 7.3 | 12.5 | RTN | 7.4 | 8.2 |
| 6 | 4-8-8 | SmoothQuant | 9.1 | 13.7 | SmoothQuant | 8.8 | 12.5 | SmoothQuant | 8.7 | 9.8 |
| 7 | 4-8-8 | LLM-QAT | **7.5** | **11.2** | LLM-QAT | **6.8** | **10.0** | LLM-QAT | **6.9** | **7.5** |
| 8 | 4-6-16 | RTN | 10.5 | 20.0 | RTN | 11.3 | 32.7 | RTN | 11.4 | 15.4 |
| 9 | 4-6-16 | SmoothQuant | 9.9 | 14.7 | SmoothQuant | 9.1 | 13.6 | SmoothQuant | 8.7 | 12.5 |
| 10 | 4-6-16 | LLM-QAT | **7.7** | **10.8** | LLM-QAT | **7.1** | **10.5** | LLM-QAT | **7.3** | **7.9** |
| 11 | 4-8-16 | RTN | 8.6 | 14.0 | RTN | 7.5 | 12.5 | RTN | 7.4 | 8.2 |
| 12 | 4-8-16 | SmoothQuant | 9.1 | 13.7 | SmoothQuant | 8.7 | 12.6 | SmoothQuant | 8.7 | 9.8 |
| 13 | 4-8-16 | LLM-QAT | **7.4** | **10.9** | LLM-QAT | **6.8** | **10.0** | LLM-QAT | **6.9** | **7.5** |
| 14 | 4-16-16 | RTN | 8.5 | 14.4 | RTN | 7.3 | 11.9 | RTN | 7.0 | 7.7 |
| 15 | 4-16-16 | GPTQ | 8.4 | 17.4 | GPTQ | 6.8 | 10.7 | GPTQ | **6.2** | 7.9 |
| 16 | 4-16-16 | LLM-QAT | **7.4** | **10.9** | LLM-QAT | **6.5** | **9.6** | LLM-QAT | 6.5 | **7.3** |
| 17 | 8-8-4 | RTN | 42.1 | 105.1 | RTN | 15.4 | 43.4 | RTN | 7.0 | 7.8 |
| 18 | 8-8-4 | SmoothQuant | 30.8 | 77.9 | SmoothQuant | 13.9 | 40.9 | SmoothQuant | **6.7** | 7.5 |
| 19 | 8-8-4 | LLM-QAT | **7.6** | **10.2** | LLM-QAT | **7.5** | **11.3** | LLM-QAT | 6.8 | **7.4** |
| 20 | 8-8-8 | RTN | 7.1 | 10.7 | RTN | 6.6 | 10.0 | RTN | 6.3 | 7.3 |
| 21 | 8-8-8 | SmoothQuant | **7.0** | 10.5 | SmoothQuant | **6.5** | 9.8 | SmoothQuant | **6.1** | **7.1** |
| 22 | 8-8-8 | LLM-QAT | **7.0** | **10.3** | LLM-QAT | 7.0 | **9.4** | LLM-QAT | 6.3 | **7.1** |
| 23 | 8-8-16 | RTN | 7.3 | 10.7 | RTN | 6.8 | 10.1 | RTN | 6.3 | 7.3 |
| 24 | 8-8-16 | SmoothQuant | **7.0** | 10.5 | SmoothQuant | **6.5** | 9.7 | SmoothQuant | **6.1** | **7.1** |
| 25 | 8-8-16 | LLM-QAT | **7.0** | **10.3** | LLM-QAT | **6.5** | **9.5** | LLM-QAT | 6.3 | **7.1** |

and higher throughput in practice. This also holds for an 8-bit 13B model compared with a 16-bit 7B model. Furthermore, 4-bit models quantized using LLM-QAT should be preferred over 8-bit models of similar size. For instance a 4-8-4 LLM-QAT 30B outperforms an 8-bit LLaMA-13B, and a 4-8-8 LLM-QAT 13B is better than an 8-bit LLaMA-7B. As a result, we recommend 4-bit LLM-QAT models for the best efficiency-accuracy tradeoff.

## 3.3 ABLATION

We conduct the ablation study regarding the data choice, quantization methods, and knowledge distillation methods in Section 3.3.1, Section 3.3.2 and Section 3.3.2, respectively. We report both the perplexity scores on WikiText2 (Merity et al., 2016)/C4 (Raffel et al., 2020) datasets and the performance on zero-shot common sense reasoning tasks.

### 3.3.1 DATA CHOICE

In Table 3, we observe that WikiText (Merity et al., 2016), which is constructed using text extracted from Wikipedia, does not encompass all the information utilized during pre-training. Consequently, a model fine-tuned solely on WikiText tends to overfit on this specific dataset and struggles to generalize well to other datasets. On the other hand, the Crawled Corpus (C4) dataset (Raffel et al., 2020) comprises hundreds of gigabytes of clean English text collected from the web. Fine-tuning the model on C4 yields reasonable transfer accuracy when evaluated on the WikiText dataset. However, it exhibits poor accuracy when tasked with zero-shot inference tasks.

Compared to the existing data, the model fine-tuned on generated data demonstrates superior generalizability, particularly in zero-shot tasks. Moreover, the data generated through sampling from the distribution exhibits greater diversity compared to the data generated without sampling. This enhanced diversity leads to significantly improved performance across all tasks.

### 3.3.2 QUANTIZATION FUNCTION

We compare the no-clipping quantization method with clipping-based methods in Table 4. Following the practice in previous works (Liu et al., 2022b; 2023), we use StatsQ (Liu et al., 2022a), a statistically-calculated scaling factor for clipping-based weight quantization and LSQ (Esser et al., 2019), the learnable scaling factor for clipping-based activation quantization. However, our findings indicate that these two state-of-the-art clipping-based quantization methods do not surpass the performance achieved by the MinMax non-clipping method. This observation reinforces the argument that preserving the outliers is critical to the performance of large language models.

Table 3: Effects of the finetuning data to the performance in downstream tasks. We use 4-bit weight 6-bit activation LLaMA-7B for the experiments. We test three strategies for data generation. Generated data[1] refers to always picking the top-1 candidate without sampling. Generated data[2] refers to sampling the next token from the distribution. Generated data[3] refers to first 3~5 tokens are generated with deterministic selection while the rest are stochastically sampled from the distribution.

| Finetuning Data | C4 ($\downarrow$) | Wiki2 ($\downarrow$) | BoolQ ($\uparrow$) | PIQA ($\uparrow$) | SIQA ($\uparrow$) | HellaSwag ($\uparrow$) | WinoGrande ($\uparrow$) | ARC-e ($\uparrow$) | ARC-c ($\uparrow$) | OBQA ($\uparrow$) | Avg. ($\uparrow$) |
|---|---|---|---|---|---|---|---|---|---|---|---|
| 1 (Pretrained Model) | 7.2 | 10.7 | 76.8 | 79.3 | 48.6 | 76.1 | 70.0 | 73.0 | 48.0 | 57.6 | 66.2 |
| 2 wiki2 | 10.1 | 5.5 | 46.9 | 74.3 | 45.2 | 72.4 | 65.7 | 67.2 | 45.0 | 47.8 | 58.1 |
| 3 wiki103 | 9.6 | **5.2** | 45.9 | 74.4 | 46.4 | 71.4 | 66.1 | 67.5 | 46.3 | 49.8 | 58.5 |
| 4 c4 | 7.8 | 11.3 | 61.7 | 77.7 | 48.8 | 73.2 | 67.2 | 67.8 | 43.6 | 52.2 | 61.5 |
| 5 Generated data[1] | 8.0 | 11.4 | 60.0 | 77.1 | 48.1 | 72.3 | 65.7 | 67.4 | 44.2 | 49.8 | 60.6 |
| 6 Generated data[2] | 7.7 | 11.5 | 70.9 | 76.1 | 47.9 | 72.2 | 66.9 | 69.3 | 46.4 | 53.6 | 62.9 |
| 7 Generated data[3] | **7.7** | 10.8 | 72.9 | 76.8 | 47.9 | 72.4 | 68.3 | 68.8 | 44.2 | 53.2 | **63.1** |

Table 4: Ablation study on the effects of the quantization methods on LLaMA-7B model. The quantization level is set to 4-bit weight and 8-bit activation.

| Weight | Activation | C4 ($\downarrow$) | Wiki2 ($\downarrow$) | BoolQ ($\uparrow$) | PIQA ($\uparrow$) | SIQA ($\uparrow$) | HellaSwag ($\uparrow$) | WinoGrande ($\uparrow$) | ARC-e ($\uparrow$) | ARC-c ($\uparrow$) | OBQA ($\uparrow$) | Avg. ($\uparrow$) |
|---|---|---|---|---|---|---|---|---|---|---|---|---|
| 1 (Pretrained Model) | | 7.2 | 10.7 | 76.8 | 79.3 | 48.6 | 76.1 | 70.0 | 73.0 | 48.0 | 57.6 | 66.2 |
| 2 Clipping | Clipping | 9.0 | 11.9 | 64.9 | 66.8 | 43.6 | 63.5 | 56.1 | 51.0 | 31.4 | 33.8 | 51.4 |
| 3 MinMax | Clipping | 9.4 | 12.8 | 63.5 | 62.4 | 42.4 | 61.2 | 52.9 | 45.6 | 29.6 | 33.8 | 48.9 |
| 4 Clipping | MinMax | 8.2 | 11.0 | 71.7 | 75.1 | 43.7 | 69.5 | 58.9 | 62.6 | 35.2 | 37.8 | 56.8 |
| 5 MinMax | MinMax | **7.4** | **10.9** | 74.8 | 77.8 | 48.6 | 73.6 | 69.0 | 69.7 | 45.8 | 55.8 | **64.4** |
| 6 Asym | Asym | 7.3 | 10.4 | 75.0 | 78.4 | 48.0 | 73.9 | 69.3 | 71.9 | 45.7 | 52.6 | 64.3 |
| 7 Sym | Asym | 7.4 | 11.0 | 72.7 | 77.9 | 48.8 | 73.3 | 67.9 | 69.2 | 45.2 | 56.0 | 63.9 |
| 8 Asym | Sym | 7.4 | 10.9 | 73.3 | 78.4 | 48.0 | 73.9 | 68.9 | 71.4 | 46.4 | 54.0 | 64.3 |
| 9 Sym | Sym | **7.4** | **10.9** | 74.8 | 77.8 | 48.6 | 73.6 | 69.0 | 69.7 | 45.8 | 55.8 | **64.4** |

Furthermore, we observe that for LLaMA models, the activations and weights exhibit predominantly symmetric distributions, which makes using symmetric quantizers the best choice. It is important to note, however, that this conclusion may not hold true for other large language models, especially those incorporating GeLU layers.

### 3.3.3 KNOWLEDGE DISTILLATION

Table 5 shows that different knowledge distillation methods have a significant impact on the final accuracy of fine-tuned models. Notably, utilizing the next token alone as the label is sub-optimal due to the inherent randomness and noise introduced by sampling from a distribution of candidates during the generation process. In contrast, logit distillation, which utilizes the complete logit distribution prediction from the teacher model, leads to superior performance of fine-tuned models compared to label-based training approaches. Interestingly, we have observed that incorporating attention distillation or hidden layer distillation actually hampers the performance. Consequently, we exclusively employ logit distillation in all our experiments.

### 3.4 TRAINING COST ANALYSIS

We use NVIDIA A100-PG509 40GB for data generation. On average, it takes 36 seconds to generate one example with a generation length up to 2048, using a batch size of 1. We use 16 A100 GPUs for data generation and it allows us to generate 100k training examples in ~2.5 days. For

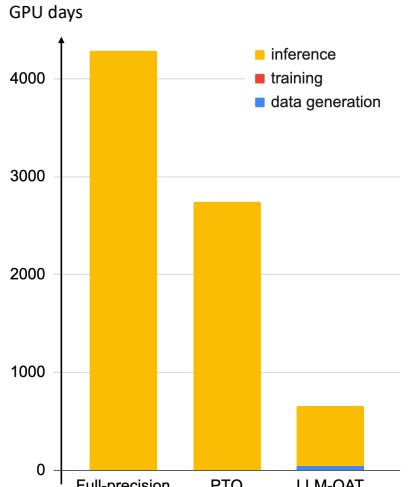

Figure 4: Total computation cost over 5 million inferences.

fine-tuning, it takes ~0.7 days ~0.8 days, and ~1.9 days to fine-tune the 7B-model and, 13B model and 30B model, respectively, with 8 A100 80G GPUs and batch size 1 per GPU on 100k generated examples. This training cost is substantially larger than PTQ, which takes 0.1 days to train. However these costs are all insignificant when amortized over the cost of inference over millions of requests.

Table 5: Ablation study on the knowledge distillation choices on LLaMA-7B model with generated data. The quantization level is set to 4-bit weight and 6-bit activation.

| Method | C4 (↓) | Wiki2 (↓) | BoolQ (↑) | PIQA (↑) | SIQA (↑) | HellaSwag (↑) | WinoGrande (↑) | ARC-e (↑) | ARC-c (↑) | OBQA (↑) | Avg. (↑) |
|---|---|---|---|---|---|---|---|---|---|---|---|
| 1 (Pretrained Model) | 7.2 | 10.4 | 76.8 | 79.3 | 48.6 | 76.1 | 70.0 | 73.0 | 48.0 | 57.6 | 66.2 |
| 2 Label | 8.1 | 11.9 | 69.4 | 77.3 | 48.7 | 72.1 | 67.1 | 67.6 | 45.4 | 51.4 | 62.4 |
| 3 Label + Attention | 8.8 | 18.6 | 70.2 | 75.3 | 47.6 | 68.9 | 67.2 | 65.6 | 42.6 | 51.2 | 61.1 |
| 4 Label + Hidden | 10.9 | 16.2 | 61.0 | 53.5 | 41.1 | 32.6 | 50.2 | 25.8 | 23.1 | 25.0 | 37.7 |
| 5 Label + Logits | 7.8 | 11.0 | 70.8 | 77.3 | 48.3 | 72.5 | 66.7 | 68.2 | 46.5 | 55.4 | **63.2** |
| 6 **Logits** | **7.7** | **10.8** | 72.9 | 76.8 | 47.9 | 72.4 | 68.3 | 68.8 | 44.2 | 53.2 | 63.1 |
| 7 Logits + Attention | 7.9 | 12.2 | 73.2 | 74.6 | 47.2 | 69.1 | 65.1 | 64.8 | 42.1 | 52.8 | 61.1 |
| 8 Logits + Hidden | 22.3 | 52.6 | 38.0 | 50.4 | 38.6 | 25.6 | 50.5 | 26.3 | 24.3 | 25.8 | 34.9 |
| 9 Logits + Hidden + Attention | 21.9 | 46.0 | 55.0 | 47.8 | 39.0 | 33.4 | 48.5 | 29.7 | 26.4 | 25.8 | 38.2 |

For instance, over 5 million inferences, a full precision 30B model would take 4280 A100 GPU days, while PTQ 8-8-8 would take 2743 days, and a comparable accuracy 4-8-8 LLM-QAT model would take 586 days[1]. These costs are pictured in Figure 4.

# 4 RELATED WORKS

**Quantization** Neural network quantization is proved to be a valuable tool in compressing model size and reducing storage consumption. Classic quantization methods, such as MinMax quantization (Jacob et al., 2018; Krishnamoorthi, 2018), Learned step-size quantization (Esser et al., 2019), PACT (Choi et al., 2018), N2UQ (Liu et al., 2022a) and etc, have primarily been developed for convolutional neural networks. While several recent works have explored language model compression, they are mostly focused on smaller models (Zafrir et al., 2019; Fan et al., 2020; Shen et al., 2020b; Zadeh et al., 2020; Bai et al., 2021; Qin et al., 2021; Liu et al., 2022b) like BERT (Devlin et al., 2019) or BART (Lewis et al., 2019). For large language models (LLMs), the available quantization methods are mostly limited to post-training quantization (Xiao et al., 2022; Yao et al., 2022; Frantar et al., 2022; Sheng et al., 2023a), due to the lack of accessible training data or the prohibitive resource requirements for fine-tuning on the entire pre-training dataset. To the best of our knowledge, no previous work has addressed the specific challenge of quantization-aware training for LLMs. The compression capabilities of state-of-the-art PTQ methods are confined to W8A8 (Xiao et al., 2022) or W4A16 (Frantar et al., 2022). A recent work introduced floating-point quantization to enable W4A8 quantization (Wu et al., 2023), however, this approach necessitates or hardware customization for floating-point computation. In contrast, LLM-QAT attains a level of accuracy on par with full-precision models with simple W4A8 integer quantization.

**Data generation** Data generation for QAT remains a relatively unexplored field of research. While there are several works in the vision domain fine-tuning student networks (Yin et al., 2020; Liu et al., 2022c; Cai et al., 2020) using noise to data generation from pre-trained teacher models, these methods mainly focus on image data. In language domain, a few previous work use human-defined prompts to elicit responses from GPT models for fine-tuning. For example, Vicuna (Zheng et al., 2023) utilized user-uploaded ShareGPT data for instruction fine-tuning, while Alpaca (Taori et al., 2023) relied on predefined human prompts with careful balance in each category to ensure diversity. In contrast, our methods eliminates the need for human prompts or user data. A single random initial token allows LLMs to autonomously generate data suitable for QAT finetuning. To the best of our knowledge, this is not studied in existing literature.

# 5 CONCLUSION AND LIMITATIONS

We proposed data-free quantization-aware training for LLMs and showed accurate, 4-bit quantization is possible using this technique. Given the generality of the training-data-agnostic distillation method, and the growing cost of LLM deployments, we expect our method to have wide applicability. For instance, the method could also be used for models trained in several stages, e.g. with instruction tuning or reinforcement learning (Ouyang et al., 2022). We leave this investigation to future work.

---

[1]8-bit weight 8-bit activation quantization results in 1.56x speedup, and 4-bit weight 8-bit activation quantization achieves 7.3x speedup according to Xiao et al. (2022); Bai et al. (2022)

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

## A APPENDIX

### A.1 FEW-SHOT EVALUATION RESULTS

Table 6 presents the few-shot performance of the quantized model on the MMLU (Hendrycks et al., 2020) and TriviaQA (Joshi et al., 2017) benchmarks.

Table 6: 5-shot few-shot exact match performance on the TriviaQA dataset and 5-shot accuracy on Massive Multitask Language Understanding (MMLU) dataset .

| | Method | #Bits | Size (GB) | MMLU Humanities (↑) | STEM (↑) | Social Sciences (↑) | Other (↑) | Average (↑) | TriviaQA (↑) |
|---|---|---|---|---|---|---|---|---|---|
| 1 | LLaMA-7B | 16-16-16 | 12.6 | 33.5 | 30.6 | 38.4 | 39.1 | 35.2 | 57.0 |
| 2 | RTN | 4-8-4 | 3.5 | 23.9 | 26.8 | 26.5 | 24.4 | 25.2 | 0.3 |
| 3 | SmoothQuant | 4-8-4 | 3.5 | 24.3 | 27.5 | 26.2 | 24.6 | 25.5 | 3.9 |
| 4 | LLM-QAT | 4-8-4 | 3.5 | 25.6 | 24.3 | 24.0 | 27.8 | **25.5** | **42.6** |
| 5 | RTN | 4-8-8 | 3.5 | 30.1 | 25.6 | 27.5 | 32.5 | 29.1 | 44.5 |
| 6 | SmoothQuant | 4-8-8 | 3.5 | 27.1 | 28.9 | 28.0 | 31.9 | 28.7 | 39.6 |
| 7 | LLM-QAT | 4-8-8 | 3.5 | 30.0 | 27.4 | 28.4 | 34.2 | **30.0** | **50.8** |
| 8 | RTN | 4-6-16 | 3.5 | 27.0 | 26.0 | 25.8 | 27.0 | 26.5 | 36.0 |
| 9 | SmoothQuant | 4-6-16 | 3.5 | 26.2 | 27.0 | 27.5 | 29.9 | 27.5 | 36.2 |
| 10 | LLM-QAT | 4-6-16 | 3.5 | 28.9 | 27.3 | 31.6 | 33.0 | **30.0** | **49.0** |
| 11 | RTN | 4-8-16 | 3.5 | 30.2 | 25.9 | 26.8 | 32.0 | 28.9 | 44.9 |
| 12 | SmoothQuant | 4-8-16 | 3.5 | 26.9 | 28.6 | 29.6 | 32.0 | 29.0 | 40.0 |
| 13 | LLM-QAT | 4-8-16 | 3.5 | 30.3 | 28.1 | 30.3 | 34.5 | **30.8** | **50.8** |
| 14 | RTN | 8-8-4 | 6.5 | 24.2 | 27.3 | 25.8 | 24.5 | 25.3 | 14.8 |
| 15 | SmoothQuant | 8-8-4 | 6.5 | 24.4 | 26.4 | 25.6 | 24.2 | 25.1 | 32.8 |
| 16 | LLM-QAT | 8-8-4 | 6.5 | 28.3 | 25.5 | 28.7 | 30.4 | **28.2** | **46.2** |
| 17 | RTN | 8-8-8 | 6.5 | 34.3 | 31.9 | 38.5 | 40.5 | **36.1** | 56.6 |
| 18 | SmoothQuant | 8-8-8 | 6.5 | 33.2 | 31.5 | 38.5 | 38.9 | 35.3 | **56.7** |
| 19 | LLM-QAT | 8-8-8 | 6.5 | 32.9 | 29.7 | 37.9 | 37.9 | 34.4 | 56.1 |
| 20 | RTN | 8-8-16 | 6.5 | 34.4 | 31.8 | 39.3 | 39.9 | **36.1** | 56.6 |
| 21 | SmoothQuant | 8-8-16 | 6.5 | 33.0 | 30.5 | 38.7 | 38.8 | 35.0 | **56.8** |
| 22 | LLM-QAT | 8-8-16 | 6.5 | 32.2 | 29.4 | 37.0 | 37.6 | 33.8 | 56.1 |
| 23 | LLaMA-13B | 16-16-16 | 24.2 | 44.4 | 36.2 | 54.3 | 53.3 | 46.7 | 63.7 |
| 24 | RTN | 4-8-4 | 6.5 | 25.5 | 24.9 | 24.3 | 26.5 | 25.3 | 22.2 |
| 25 | SmoothQuant | 4-8-4 | 6.5 | 25.6 | 22.8 | 23.4 | 26.4 | 24.7 | 32.7 |
| 26 | LLM-QAT | 4-8-4 | 6.5 | 29.4 | 28.5 | 31.9 | 35.8 | **31.1** | **54.3** |
| 27 | RTN | 4-8-8 | 6.5 | 38.3 | 32.7 | 45.3 | 46.4 | 40.4 | 57.9 |
| 28 | SmoothQuant | 4-8-8 | 6.5 | 30.9 | 28.6 | 33.4 | 37.1 | 32.3 | 46.6 |
| 29 | LLM-QAT | 4-8-8 | 6.5 | 38.7 | 32.8 | 47.1 | 47.7 | **41.2** | **59.3** |
| 30 | RTN | 4-6-16 | 6.5 | 28.5 | 27.8 | 29.5 | 32.0 | 29.3 | 39.6 |
| 31 | SmoothQuant | 4-6-16 | 6.5 | 30.3 | 29.6 | 33.5 | 37.1 | 32.4 | 44.8 |
| 32 | LLM-QAT | 4-6-16 | 6.5 | 37.4 | 33.4 | 45.1 | 46.0 | **40.1** | **57.7** |
| 33 | RTN | 4-8-16 | 6.5 | 38.7 | 32.6 | 45.2 | 45.8 | 40.3 | 57.9 |
| 34 | SmoothQuant | 4-8-16 | 6.5 | 30.3 | 27.8 | 34.3 | 37.5 | 32.2 | 46.6 |
| 35 | LLM-QAT | 4-8-16 | 6.5 | 40.1 | 32.4 | 47.6 | 48.0 | **41.8** | **59.8** |
| 36 | RTN | 8-8-4 | 12.4 | 27.8 | 26.2 | 27.0 | 29.6 | 27.6 | 44.3 |
| 37 | SmoothQuant | 8-8-4 | 12.4 | 27.8 | 28.1 | 28.6 | 32.3 | 29.1 | 49.6 |
| 38 | LLM-QAT | 8-8-4 | 12.4 | 34.1 | 29.3 | 38.7 | 40.7 | **35.5** | **58.8** |
| 39 | RTN | 8-8-8 | 12.4 | 44.2 | 35.6 | 52.2 | 52.5 | 45.9 | 62.9 |
| 40 | SmoothQuant | 8-8-8 | 12.4 | 44.5 | 36.1 | 53.5 | 53.3 | **46.6** | **63.4** |
| 41 | LLM-QAT | 8-8-8 | 12.4 | 43.5 | 36.1 | 52.6 | 52.5 | 45.8 | 63.3 |
| 42 | RTN | 8-8-16 | 12.4 | 44.3 | 34.9 | 51.7 | 53.0 | 45.7 | 63.1 |
| 43 | SmoothQuant | 8-8-16 | 12.4 | 44.5 | 36.4 | 53.7 | 53.4 | **46.7** | **63.4** |
| 44 | LLM-QAT | 8-8-16 | 12.4 | 43.6 | 36.1 | 53.8 | 53.2 | 46.3 | **63.4** |
| 23 | LLaMA-30B | 16-16-16 | 60.6 | 55.8 | 46.0 | 66.7 | 63.4 | 57.8 | 69.9 |
| 46 | RTN | 4-8-4 | 15.7 | 24.4 | 26.2 | 27.2 | 26.4 | 25.9 | 19.2 |
| 47 | SmoothQuant | 4-8-4 | 15.7 | 23.9 | 27.5 | 23.2 | 24.1 | 24.6 | 7.5 |
| 48 | LLM-QAT | 4-8-4 | 15.7 | 47.6 | 40.4 | 55.9 | 54.5 | **49.3** | **63.5** |
| 49 | RTN | 4-8-8 | 15.7 | 51.0 | 43.6 | 62.2 | 60.6 | 53.9 | **66.8** |
| 50 | SmoothQuant | 4-8-8 | 15.7 | 35.2 | 35.1 | 46.9 | 45.2 | 40.0 | 57.9 |
| 51 | LLM-QAT | 4-8-8 | 15.7 | 52.2 | 44.3 | 61.4 | 61.0 | **54.4** | 65.9 |
| 52 | RTN | 4-6-16 | 15.7 | 29.5 | 31.3 | 32.1 | 36.2 | 32.0 | 39.3 |
| 53 | SmoothQuant | 4-6-16 | 15.7 | 31.6 | 34.3 | 43.4 | 42.3 | 37.2 | 56.7 |
| 54 | LLM-QAT | 4-6-16 | 15.7 | 47.7 | 41.7 | 58.9 | 57.5 | **51.0** | **64.2** |
| 55 | RTN | 4-8-16 | 15.7 | 50.9 | 44.0 | 62.8 | 61.3 | 54.2 | **67.1** |
| 56 | SmoothQuant | 4-8-16 | 15.7 | 35.6 | 36.2 | 48.6 | 45.7 | 40.8 | 58.5 |
| 57 | LLM-QAT | 4-8-16 | 15.7 | 52.8 | 44.4 | 63.6 | 61.2 | **55.1** | **67.1** |
| 58 | RTN | 8-8-4 | 30.7 | 26.1 | 27.6 | 28.6 | 29.0 | 27.6 | 30.2 |
| 59 | SmoothQuant | 8-8-4 | 30.7 | 27.9 | 29.1 | 31.7 | 33.1 | 30.1 | 38.9 |
| 60 | LLM-QAT | 8-8-4 | 30.7 | 49.7 | 42.2 | 60.8 | 59.7 | **52.7** | **67.9** |
| 61 | RTN | 8-8-8 | 30.7 | 55.6 | 45.8 | 66.3 | 63.4 | 57.5 | **70.4** |
| 62 | SmoothQuant | 8-8-8 | 30.7 | 56.0 | 46.0 | 67.3 | 64.1 | 58.0 | 70.2 |
| 63 | LLM-QAT | 8-8-8 | 30.7 | 56.5 | 47.7 | 66.9 | 64.2 | **58.5** | 69.4 |
| 64 | RTN | 8-8-16 | 30.7 | 56.3 | 45.6 | 66.8 | 63.7 | 57.8 | **70.3** |
| 65 | SmoothQuant | 8-8-16 | 30.7 | 56.0 | 46.7 | 67.5 | 63.8 | **58.2** | **70.3** |
| 66 | LLM-QAT | 8-8-16 | 30.7 | 54.9 | 45.9 | 66.7 | 63.6 | 57.4 | 70.0 |

Table 7: Explore 4-bit weight 4-bit activation quantization with `LLM-QAT`.

| Method | #Bits | BoolQ (↑) | PIQA (↑) | SIQA (↑) | HellaSwag (↑) | WinoGrande (↑) | ARC-e (↑) | ARC-c (↑) | OBQA (↑) | Avg. (↑) |
|---|---|---|---|---|---|---|---|---|---|---|
| 1 LLaMA-7B | 16-16-16 | 76.8 | 79.3 | 48.6 | 76.1 | 70.0 | 73.0 | 48.0 | 57.6 | 66.2 |
| 2 RTN | 4-4-16 | 51.3 | 49.8 | 36.9 | 26.2 | 47.9 | 25.7 | 24.5 | 31.2 | 36.7 |
| 3 SmoothQuant | 4-4-16 | 54.1 | 62.8 | 41.8 | 41.5 | 52.6 | 50.6 | 32.9 | 36.4 | 46.6 |
| 4 LLM-QAT | 4-4-16 | 57.9 | 47.5 | 39.9 | 25.8 | 47.6 | 27.2 | 25.8 | 29.4 | 37.6 |
| 5 LLM-QAT + SmoothQuant | 4-4-16 | 63.5 | 64.3 | 41.8 | 55.6 | 52.9 | 50.3 | 30.2 | 35.0 | 49.2 |
| 6 LLM-QAT + group-wise quant (4 channel per group) | 4-4-16 | 65.5 | 74.0 | 47.7 | 68.1 | 65.4 | 66.5 | 43.9 | 52.4 | 60.4 |
| 7 LLM-QAT + group-wise quant (1 channel per group) | 4-4-16 | 69.1 | 75.5 | 47.4 | 70.5 | 66.9 | 67.6 | 46.8 | 50.2 | 61.7 |
| 8 RTN | 4-4-4 | 50.2 | 50.5 | 37.1 | 26.0 | 49.6 | 26.1 | 24.4 | 28.6 | 36.6 |
| 9 SmoothQuant | 4-4-4 | 49.1 | 49.8 | 39.1 | 27.4 | 48.0 | 30.4 | 25.8 | 29.2 | 37.4 |
| 10 LLM-QAT | 4-4-4 | 61.3 | 51.5 | 39.2 | 31.1 | 51.9 | 27.9 | 23.9 | 29.4 | 39.5 |
| 11 LLM-QAT + SmoothQuant | 4-4-4 | 62.4 | 55.9 | 40.9 | 47.8 | 50.6 | 35.5 | 26.4 | 34.6 | 44.3 |
| 12 LLM-QAT + group-wise quant (4 channel per group) | 4-4-4 | 60.3 | 66.3 | 45.4 | 56.8 | 57.1 | 54.9 | 34.1 | 38.6 | 51.7 |
| 13 LLM-QAT + group-wise quant (1 channel per group) | 4-4-4 | 67.9 | 74.2 | 46.6 | 66.8 | 59.4 | 63.9 | 41.3 | 48.8 | 58.6 |

## A.2 EXPLORING THE LIMITS: 4-BIT WEIGHT 4-BIT ACTIVATION QUANTIZATION

We further explore the lower-bit quantization of 4-bit weight and 4-bit activation (W4A4). The results show that the W4A4 quantization is challenging for LLMs. Post-training quantization sees $\sim 30$ points degradation. Adding LLM-QAT together with smoothquant can recover 12.5 points and 7.7 points for W4A4KV16 and W4A4KV4 settings, respectively.

Furthermore, we delved into the potential of combining the group-wise quantization (Shen et al., 2020a; Sheng et al., 2023a) with LLM-QAT. The results in Table 7 unveiled that group-wise quantization with 1-channel per group managed to achieve less than 8 points accuracy drop on the W4A4KV4 setup compared to full-precision, which was infeasible in any of the previous works. Nevertheless, it is important to note that implementing this group-wise quantization method may necessitate specialized kernel design to fully realize its potential for actual speedup (Shen et al., 2020a; Cai et al., 2020).

## A.3 EVALUATION BENCHMARKS

### A.3.1 ZERO-SHOT COMMON SENSE REASONING TASKS

**BoolQ** (Clark et al., 2019) is a reading comprehension dataset of naturally occurring yes/no questions. Each example consists of a question (Q), an excerpt from a passage (P), and an answer (A) with an explanation added for clarity.

**PIQA** (Bisk et al., 2020), short for Physical Interaction: Question Answering, is a benchmark for evaluating and studying physical commonsense understanding in natural language models.

**SIQA** (Sap et al., 2019) aims to measure the social and emotional intelligence of computational models through multiple choice question answering (QA).

**HellaSwag** (Zellers et al., 2019) is a benchmark for physically situated commonsense natural language inference. It consists the four-way multiple-choice problems that are trivial for humans (> 95% accuracy), but challenging for the language models.

**WinoGrande** (Sakaguchi et al., 2021) is a benchmark for commonsense reasoning. It comprises a set of 273 expert-crafted pronoun resolution problems originally designed to be unsolvable for statistical models that rely on selectional preferences or word associations.

**ARC** (Clark et al., 2018), the AI2 Reasoning Challenge, contains a collection of 7787 natural science questions. It is partitioned into a Challenge Set and an Easy Set, where the Challenge Set contains only questions answered incorrectly by both a retrieval-based algorithm and a word co-occurrence algorithm.

**OBQA** (Mihaylov et al., 2018) is a dataset of about 6000 questions for open book question answering. The task focuses on the challenge of combining a corpus of provided science facts (open book) with external broad common knowledge.

### A.3.2 FEW-SHOT TASKS

**TriviaQA** (Joshi et al., 2017) is a closed-book question answering benchmark. It contains over 650K question-answer evidence triples, that are derived by combining 95K Trivia enthusiast authored question-answer pairs with on average six supporting evidence documents per question.

**MMLU** (Hendrycks et al., 2020), the Massive Multitask Language Understanding(MMLU) benchmark (Hendrycks et al., 2020), consists of multiple choice questions covering various domains of knowledge, including humanities, STEM and social sciences.

## A.4 Generation Tasks

**WikiText2** (Merity et al., 2016) is a collection of over 100 million tokens extracted from the set of verified Good and Featured articles on Wikipedia.

**C4** (Raffel et al., 2020), abbreviate for Colossal Clean Crawled Corpus. Common Crawl[2] is a publicly-available web archive that provides "web extracted text" by removing markup and other non-text content from the scraped HTML files. C4 filters Common Crawl's web-extracted text and produces a collection of text that comprises clean and natural English text.

## A.5 Generated data from LLaMA-7B

In this section, we show some examples of the generated data obtained through the next token generation with LLaMA-7B model.

---

\nSincerely, I\u2019m sending for your help right now.\nI\u2019m currently facing a rather challenging situation.\nMy wife and me have been living and working in the same country for a couple of years.\nHowever, after we\u2019ve seen a couple of movies together, we found out that we have different tastes in movies.\nMy wife is always watching melancholy (soporific, slow-moving) drama movies, while I prefer thrillers or action movies.\nIt would be great if we could both keep our personal wishes, yet make sure we have something to discuss about afterwards.\nDo you think there\u2019s a nice web site, a movie club, or so, which helps us choose movies to watch as a couple?\nPlease, please help me out of this situation!\nA: Thanks for asking for my help. If you can, please do watch some movies together before choosing the movies that are really suitable for each of you.\nMovies have different styles of story telling, and some can be slow-moving. It is up to the audience to decide what kind of movies they want to see. You are just the same as other people who might prefer watching some movies at home while the other one loves to watch movies at a movie theater.\nIf there exist some websites or movie clubs, why don\u2019t you try taking a look at it. However, it is up to you to find out which movies are suitable for your couple, so the movie club, or the website, can only give some ideas for your reference.\nSincerely, hope my answer could help you with your issue.\n- I am new to ALT life and have been living here for three month. We would love to join some clubs and have asked many people with no success. I don't understand why some people are unwilling to help.\n- It's always best to ask people at the local supermarket or caf\u00e9 for any events coming up in the area. The best way to find out is to ask.\n- I'd rather not answer at this stage.\n- I like this place, and I'd like to stay here, but I am sure there is more to see and do, so hopefully if I try hard enough I will be able to find out more.\n- I think it's a bit difficult living away from home, but having family and friends around helps a lot.\n- We'd like to play baseball, is there any baseball or softball club, we are new to your country but quite enthusiastic.\n- We have a large number of volunteer groups that actively help the local community, and the local government sets up new programmes all the time."

---

[2]http://commoncrawl.org/

In the mid-20th century, there was growing awareness that in a large number of industrial countries the population was aging. This awareness, along with advances in social security in many countries and a sense of urgency to avoid future generations of poor, has led to social policy being implemented that targets the elderly. The elderly often have a special status and enjoy particular social benefits. Such benefits include a higher age entitlement threshold, higher pensions, early-retirement benefits, employment quotas, and free or subsidized health care. In addition, in some countries the elderly are exempt from federal and, in some cases, from state taxes. \n Since most governments have set limits on their ability to pay out more in social security and health care benefits to any one group, such benefits are generally reserved for the retired and the disabled. Accordingly, elderly care for the mentally disabled, infirm, and cognitively impaired has taken on particular importance, as has care for the homeless. In addition, those who are economically active can also benefit from programs, although there is the perception that the economically active are less in need of care assistance, because their families often take care of them. Elderly care is generally provided on an unpaid basis, although those in need of care may have to sell their homes because of the high cost of care.\nSee also: Economy; Old Age and Retirement Welfare Systems; Unemployment; and Unemployment and Employment Security Systems.\nBell, Ian R. S., The Growing Crisis of Old Age. London: Pall Mall Press, 1913.

When my brother returned from abroad, he went with his wife to the motherland. They had many elder brothers around there, so that we all went with them to my parents\u2019 home. I, being the youngest, went first. My brother, seeing the dowry that I brought with me, laughed. He was very proud. \u2018This girl is from my sister; she is the same as my sister. Why should I have her?\u2019 he said to them. But at that time my father was ill; he lay stretched on a bed before us. \u2018This is my sister; she is my sister. I did not give her to you; why should I allow you to kill my sister?\u2019 the father said to my brother. His mother said, \u2018I shall put this girl\u2019s hand in your hand. When you wish to give it away, it will be like your sister\u2019s.\u2019 My brother thought, \u2018There surely cannot be two sisters.\u2019 I told him, \u2018The old man will die soon. I have brought him his dowry of ten yens of silver. It is no small gift for him. If I do you the favour now, will you not have respect for the father when he is dead and gone?

a town, and the largest inland city of the ancient Aztlan (Aztatlan, Azatlan) located near the modern ruin of Santa Anas, Sonora, Mexico.\nAtsa was a settlement among the 13 Tamoanchan cities of the Nahua people that established their settlement in Northern Mexico by about 1000 A.D. The city of Atsa (its people Atsatla) belonged to the twelve tribes of Tamoanchan, which may be read in the Aztec codices as Tlascallaänd which are the Tlascalan peoples of the 13 cities or a confederation of peoples that were among the dominant rulers of the Valley of Mexico, and of western Mexico in general, during the pre-eminent period of Aztec civilization.\nAtsa as well as the other Tamoancan cities of the Aztecs, is said to be on a plain of jade and silver and to have a number of large, flat-roofed buildings. These buildings are described as being in ruins in 1519, during Hernan Cortes's first visit to Mexico. Although it is not known if the Tamoancan cities were named before or after contact with the Spaniards, it may be that the descriptions are only describing Aztec ideas of the 13 cities by its inhabitants. Atsa is said to be located at a point where a landform resembling a large lake (a mountain), was formed at some point in their past that has since dried. It is here that it is said the Atsatla people established their community.

This article appeared in the Ceylon Today on 24th January 2016.\nI have been an Anglican Priest and a member of various committees in the Ceylon Evangelical Lutheran Church (CELC) for many years. I have observed that most of the CELC clergy and Laity are engaged in one or the other form of mission; whether as pastor, deacon, Sunday School teacher, a member of the Christian Education Council (CEC), Church Council or its equivalent. But do we realise the full significance of the words of the Apostle Paul in the first Reading?\nThe readings for this Sunday continue in Paul\u2019s second letter to the church in Corinth, where he warns the Christians against the excesses of the Eucharist. He warns them not to drunkenness during the Eucharist. He also writes: \u201cDo not, for the sake of food, destroy the work of God\u201d (6:12).\nThese were the words Paul used in his letter to his beloved Church: \u201cEat anything God has created to make you healthy.\nYou should not feel guilty about eating any of these foods, but you should not eat them if they cause other people to stumble\u201d (Romans 14:14-23, CEB).\nIf we want to do God\u2019s will, we will learn what pleases him. (Eph. 5:10). If we want others to respect God\u2019s law, then we must avoid even the appearance of sin (1 Thess. 5:22). In other words, we must seek to please God in all of our actions.\nAs Catholics we celebrate the Holy Mass by taking communion of the Bread and Wine, in the name of the Father, the Son and the Holy Spirit to remind us of our call to love and serve one another.\nSt. Paul said: \u201cA little yeast spreads through the whole batch of dough\u201d (Gal. 5:9). Christians should celebrate God\u2019s love in a special way. We are called to live for the next generation. If we drink to excess at the Eucharist it ruins the lives of our children. \u201cA little yeast\u201d spreads quickly through the family: an immoral step, a broken marriage, divorce and abuse: All these are examples of the \u201clittle yeast\u201d that destroys families.\nBut there\u2019s more. \u201cBecome sober, and stop sinning. Then some apostle will not be wrongly accused of being responsible for sinning. Anyone who has been stealing must steal no longer\u201d (v. 8). Our call as Christians is to lead exemplary lives. As such, we must act in accordance with our commitment to the \u201cLove of God and of our Neighbour\u201d. We are called to work with integrity. We are called to give to society. If a person is addicted to any alcoholic beverage or to any other type of drug, it will be difficult for him or her to lead a moral life.\nIt is also said that the words of the Apostle Paul in the First Reading \u201care still being applied today\u201d!\nOur forefathers who were the followers of John Wesley the Founder of Methodism made this famous saying to describe a Methodist: \u2018 A Methodist will find his way home from the most God forsaken, lawless, brutalized and degraded part of the earth, to the humble home of his childhood or his God: To find his way when the sun sets and the sky is filled with dense shades of night; or on the banks of the Red river, amid the solitudes of the Mexican desert, or in the dreary solitude of the frozen wastes of the Arctic Ocean\u2026\u201d.\nIf we ask ourselves some important question we are compelled to ask such question whether this is true for every individual member of the churches, whether it is true for the clergy or laity.\nThe Methodist Church in the United Kingdom, for example, is known as the \u201cMother Church\u201d. In 2014, a survey revealed that 21% of the British public was unaffiliated to any religion. But 50% of the people in England and Wales were prepared to \u201ctry and become a Christian\u201d if invited, despite not going to church."

## A.6 BROADER IMPACT

We propose a model compression technique that reduces the memory footprint of large language models, enabling their deployment on embedded devices. This technique has the potential to decrease energy consumption for end users and lower costs for companies running language models at scale.

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
