# OpenReview forum: "LLM-QAT: Data-Free Quantization Aware Training for Large Language Models"
_ICLR.cc/2024/Conference — Submitted to ICLR 2024_

### Official Review · Reviewer_dtJe · 2023-10-31

**Soundness:** 2 fair
**Presentation:** 3 good
**Contribution:** 2 fair
**Rating:** 5
**Confidence:** 4

**Summary:**

The paper investigates the data-free quantization-aware training on large language models. Without the available training data, it proposes to make the LLMs generate data by themselves and study several sampling strategies. Experiments are done on a wide variety of datasets including common sense reasoning, perplexity evaluation, and MMLU. It also conducts experiments across several bit-widths on the LLaMA model.

**Strengths:**

* Using LLMs to generate data is very natural. The paper makes a combination of it and data-free quantization, which can be a challenge for QAT.

* The paper considers several datasets to evaluate, especially include the MMLU and TriviaQA datasets in the appendix.

* The writing of the paper is clear and easy to follow.

**Weaknesses:**

* As using LLMs to generate data is not a very new thing [1][2], can the paper give more detailed analyses about the synthetic method or the synthetic results like sentence coherence and data diversity? I think this can increase the soundness of the proposed method.

  [1]. Generating Faithful Synthetic Data with Large Language Models: A Case Study in Computational Social Science

  [2]. CLASP: Few-Shot Cross-Lingual Data Augmentation for Semantic Parsing


* Experiments parts are not very solid:
  * The 4-6-16 setting of experiments is a little bit strange and unfriendly to hardware. I think the paper should move 4-4-4 experiments that take 4-bit weight and activations from the appendix to the main body.
  * I find the fifth row in Table 1 looks strange It seems that the performance of GPTQ is really slow, even slower than RTN with 4-8-16 bits.
  * Experiments can be more convincing if the paper also conducts on other structures like LLaMA-2, and others take the vanilla
   Transformer structure as LLaMA adopts a different one.
  * Experiments can be better if the paper could compare with some recent papers [3] and [4].

  [3]. Outlier Suppression+: Accurate quantization of large language models by equivalent and effective shifting and scaling

  [4]. OmniQuant: Omnidirectionally Calibrated Quantization for Large Language Models

**Questions:**

Based on my practical experience and discussion with some people in this field, we find quantizing KV cache with per-token quantization on LLaMA models will not incur much accuracy decline, even on a 4-bit case. However, results in the main table seem to give an opposite conclusion. Also, the paper claims that distribution for LLaMA models is usually symmetric. However, we find products of the output of gate and up functions in LLaMA are not so.

I do not put the above points in the weakness part temporarily because I think this requires more discussion with the authors. Others questions please check the weakness part.

---

> ### Author Response · Authors · 2023-11-23
>
> **Q1**: More discussions to the previous work.
> **A1**: Thanks for suggesting additional papers for comparison and ablation experiments. It's crucial to note that [1], [3], and [4] represent concurrent works yet to be published by the ICLR submission deadline. As per the ICLR reviewer guidelines, not comparing to these works shouldn't be grounds for rejection. The rapid evolution within the Language Model Models (LLMs) community makes it challenging to track every new development. Regarding [2], it focuses on using generated data for data augmentation in cross-lingual semantic parsing tasks. In contrast, our emphasis lies in showcasing that exclusive utilization of generated data can effectively fine-tune precise quantized LLMs. The primary focuses between the two approaches are distinct. Nonetheless, these papers are good work and we'll incorporate discussions about them in the related works section.
>
> **Q2**: The 4-6-16 setting of experiments is a little bit strange and unfriendly to hardware. I think the paper should move 4-4-4 experiments that take 4-bit weight and activations from the appendix to the main body.
> **A2**: Thanks for this suggestion. We will move the 4-4-4 experiments to the main table. it's noteworthy that W4A6 quantization can effectively reduce GPU run-time memory usage compared to W4A8. Additionally, it operates at a speed comparable to W4A8 when using dequantization.
>
>
> **Q3**: I find the fifth row in Table 1 looks strange. It seems that the performance of GPTQ is really low, even lower than RTN with 4-8-16 bits.
> **A3**: Regarding our previous experiments with GPTQ for LLaMA-7B, we found that the GPTQ is sensitive to calibration data choice. Initially, our use of C4 to calibrate all tasks wasn't optimal. Adjusting the calibration dataset to match the evaluation dataset yielded improved results for W4A16 GPTQ, as shown in the following table and will be reflected in the updated Table 1.
>
> | boolq | piqa | siqa | hellaswag | winogrande | arc_easy | arc_challenge | obqa | avg. |
> |-|-|-|-|-|-|-|-|-|
> |68.7 | 78.2 | 49.0 | 74.0 | 68.1 | 69.9 | 45.8 | 57.0 | 63.8 |
>
> **Q4**: Experiments can be more convincing if the paper also conducts on other structures like LLaMA-2, and others take the vanilla Transformer structure as LLaMA adopts a different one.
> **A4**:  Following your suggestion, we conducted experiments on the OPT-125M model, revealing that the W4A16KV16 LLM-QAT model achieves performance on par with full-precision models. Detailed results for other bit settings will be incorporated into the manuscript.
> | #bits | boolq | piqa | siqa | hellaswag | winogrande | arc_easy | arc_challenge | obqa | avg. |
> |-|-|-|-|-|-|-|-|-|-|
> | full-precision | 40.8 | 24.8 | 60.8 | 62.5 | 41.5 | 33.0 | 32.7 | 49.7 | 43.2 |
> | 4-16-16 | 40.6 | 25.1 | 58.3 | 61.1 | 40.7 | 32.8 | 32.4 | 53.8 | 43.1 |
>
> **Q5**: Based on my practical experience and discussion with some people in this field, we find quantizing KV cache with per-token quantization on LLaMA models will not incur much accuracy decline, even on a 4-bit case. However, results in the main table seem to give an opposite conclusion. Also, the paper claims that distribution for LLaMA models is usually symmetric. However, we find products of the output of gate and up functions in LLaMA are not so.
> **A5**: Regarding your experiment showing a loss-less accuracy in KV-cache quantization, we're curious if you tested the model on generation tasks that construct sequences token by token. Typically, common evaluation frameworks assess perplexity for datasets like wiki and C4, by predicting the next token based on the ground truth context, instead of the generated tokens, which is similar to the training phase. In that case, the KV cache is not used. Additionally, we confirmed that the gate function output distribution isn't symmetric-like. While symmetric quantization works well for LLaMA models, we'll modify the claim to prevent potential misinterpretation.

---

### Official Review · Reviewer_WoMH · 2023-10-31

**Soundness:** 2 fair
**Presentation:** 3 good
**Contribution:** 3 good
**Rating:** 5
**Confidence:** 4

**Summary:**

LLM-QAT paves the way for quantization-aware training (QAT) of large language models (LLMs) using cross-entropy-based logit distillation. For both weights and activations, LLM-QAT adopts the symmetric MinMax quantization format. This decision was based on an empirical analysis that took into account the outliers in LLMs. Specifically, they adopted per-token activation quantization and per-channel weight quantization. Furthermore, because the key-value cache consumes a non-negligible amount of memory, LLM-QAT quantizes the key-value cache to 8-bit or even 4-bit to increase throughput. While LLMs are sensitive to the training dataset, LLM-QAT explores training datasets for QAT that generalize well to other tasks. The empirical results of their experiments justify LLM-QAT's formulation of the quantization function and data selection. Evaluation results show that LLM-QAT offers competitive performance compared to training-free methods such as round-to-nearest (RTN) and SmoothQuant.

**Strengths:**

* Investigate a quantization format to mitigate perplexity loss, drawing from their observations about the presence of outliers in quantization-aware training (QAT).
* Provide new empirical insights on the selection of the knowledge distillation dataset and its associated loss.
* Offer comprehensive experimental results spanning various bit-widths for weight, activation, and key-value cache.
* Exhibit overall strong quality and clarity in the presentation.

**Weaknesses:**

* Compare LLM-QAT only to training-free methods such as round-to-nearest (RTN) and SmoothQuant [4].
    * Although AdaRound [1], AdaQuant [2], and FlexRound [3] are post-training quantization (PTQ) methods, they still utilize a training set (small calibration set) in a layer-wise or block-wise manner. Given that these PTQ methods adopt a similar knowledge distillation scheme (even though the loss term is very different), it would be beneficial to report results comparing LLM-QAT with these methods (at least one of them), which can be viewed as non-training-free quantization methods.
* The arguments would be more convincing with additional references and explanations for why LLM-QAT primarily focuses on the W4A8 quantization format. For instance, regarding the W8A8 integer quantization, SmoothQuant [4] showcases actual latency acceleration results.

[1] Nagel, Markus, et al. "Up or down? adaptive rounding for post-training quantization." International Conference on Machine Learning. PMLR, 2020.
[2] Hubara, Itay, et al. "Accurate post training quantization with small calibration sets." International Conference on Machine Learning. PMLR, 2021.
[3] Lee, Jung Hyun, et al. "FlexRound: Learnable Rounding based on Element-wise Division for Post-Training Quantization." International Conference on Machine Learning. PMLR, 2023.
[4] Xiao, Guangxuan, et al. "Smoothquant: Accurate and efficient post-training quantization for large language models." International Conference on Machine Learning. PMLR, 2023.

**Questions:**

* Are there experimental results available for other models?
* Do models other than LLaMA exhibit similar trends in quantization format settings?
* In Section 4, "Related Works", I kindly suggest modifying the last sentence: "A single random initial token allows LLMs to autonomously generate data suitable for QAT fine-tuning. To the best of our knowledge, this has not been studied in existing literature." LLM-QAT employs a knowledge distillation (KD) scheme for quantization-aware training (QAT), and a previous method named ZeroQuant [1] also uses the KD scheme. Even though these methods primarily focus on post-training quantization, they also utilize a training dataset for quantization. Specifically, ZeroQuant presents data-free quantization that employs random data for generative language models.
* There are typos in the manuscript, notably in sections 2.2.2 and 3.3. I suggest revising them.

[1] Yao, Zhewei, et al. "Zeroquant: Efficient and affordable post-training quantization for large-scale transformers." Advances in Neural Information Processing Systems 35 (2022): 27168-27183.

---

> ### Author Response · Authors · 2023-11-23
>
> **Q1**: Compare LLM-QAT only to training-free methods such as round-to-nearest (RTN) and SmoothQuant [4].
> **A1: The comparison to the SmoothQuant paper is extensively discussed in our main Tables (Table 1 and 2). We also compared with another SoTA PTQ method for LLMs, GPTQ.**  These two PTQ methods are the state-of-the-art PTQ methods for LLMs. In the GPTQ paper, thorough comparisons were made between their method and various Post-Training Quantization (PTQ) approaches on different language models, showcasing GPTQ's superior performance. Hence, we select the most effective PTQ method for LLMs as a benchmark for comparison with our approach. Reproducing other PTQ methods on Language Model Models (LLMs) within a short timeframe poses challenges, and we avoid presenting rushed and potentially suboptimal results for other methods.
>
> **Q2**: The arguments would be more convincing with additional references and explanations for why LLM-QAT primarily focuses on the W4A8 quantization format.
> **A2: It's worth noting that we did NOT only do experiments on W4A8; rather, we experimented with various bit settings, including W4A6, W4A16, W8A8, and W8A16, etc, detailed in in our main Tables (Table 1 and 2).** Therefore, we don’t quite agree with reviewer's comment that 'LLM-QAT primarily focuses on the W4A8 quantization format'.
>
> **Q3**: Are there experimental results available for other models?
> **A3**: Following your suggestion, we conducted experiments on the OPT-125M model, revealing that the W4A16KV16 LLM-QAT model achieves performance on par with full-precision models. Detailed results for other bit settings will be incorporated into the manuscript.
> | #bits | boolq | piqa | siqa | hellaswag | winogrande | arc_easy | arc_challenge | obqa | avg. |
> |-|-|-|-|-|-|-|-|-|-|
> | full-precision | 40.8 | 24.8 | 60.8 | 62.5 | 41.5 | 33.0 | 32.7 | 49.7 | 43.2 |
> | 4-16-16 | 40.6 | 25.1 | 58.3 | 61.1 | 40.7 | 32.8 | 32.4 | 53.8 | 43.1 |
>
> **Q4**: ZeroQuant presents data-free quantization that employs random data for generative language models.
> **A4**: We'll include a discussion on ZeroQuant in related works. We would like to emphasize two key distinctions: (1) the random tokens in ZeroQuant is not generated from the first token using LLMs, rather, they employ entirely random tokens for entire sentences, differing from our methodology. (2) the quality requirements for calibration data in PTQ are significantly less demanding compared to the quantity needed for training data in QAT.

---

### Official Review · Reviewer_hBhR · 2023-10-31

**Soundness:** 3 good
**Presentation:** 3 good
**Contribution:** 2 fair
**Rating:** 5
**Confidence:** 4

**Summary:**

The paper introduces a data free distillation approach for quantization aware training (QAT) for LLMs to address limitations related to obtaining large-scale training data and complex pre-training procedures. The paper proposes utilizing tokens generated from the generative model as the fine-tuning dataset and demonstrates that this setup works better in practice than using subsets of the original data used for training as it provides a better representation of the original data distribution. Using the data free distillation technique and existing quantization methods, the authors demonstrate better results, particularly for lower precisions, on LLaMA models of sizes 7B, 13B, and 30B than PTQ techniques from literature while quantizing weights, activations and the KV cache to lower precisions (as low as 4-bits).

**Strengths:**

The paper motivates the problem well, and quantizing large language models to lower precisions for faster inference is a very relevant research problem today.

The paper is generally well written, and includes experiments comparing various quantization techniques and also includes ablation studies on dataset choices.

This is the first application of QAT to large language models, although previous works (for example [1] have demonstrated QAT results for BERT).




[1] https://openreview.net/pdf?id=EZQnauHn-77

**Weaknesses:**

The techniques utilized in the paper (QAT using StatsQ and LSQ, MinMax quantization, knowledge distillation) have been shown to work in the quantization setting before, so the contributions are not particularly novel. Effectiveness of MinMax for language models due to outliers has also been demonstrated earlier.

While the results are generally better than PTQ techniques, specially as precision is lowered (this is usually the case for low precision scenarios - QAT performs better than PTQ), the gaps are still quite large compared to full precision to be practically useful, specially for the 4-bit case, and I am not entirely convinced from the limited ablation studies that fine-tuning on a subset of the original data would not yield comparable results. While practically useful, the algorithmic contribution in this work is weak.

**Questions:**

For the dataset ablation study, have the authors considered using a subset of each dataset and comparing that to generated data, instead of exclusively fine-tuning using a single dataset?

Perhaps using a combination of real and generated data might close the accuracy gap further?

Have the authors considered using a larger teacher model for the distillation data generation?

---

> ### Author Response · Authors · 2023-11-23
>
> **Q1**: The techniques utilized in the paper (QAT using StatsQ and LSQ, MinMax quantization, knowledge distillation) have been shown to work in the quantization setting before, so the contributions are not particularly novel. Effectiveness of MinMax for language models due to outliers has also been demonstrated earlier.
> **A1**: The primary focus of our paper lies not in the design of the quantization function but rather in presenting an innovative data generation pipeline facilitating Quantization Aware Training (QAT) for Language Model Models (LLMs) within feasible computational constraints, as detailed in Lines 32-42 of our manuscript. The previous works Vicuna [1] and Alpaca [2] stand as the most relevant works in this domain, while a distinct divergence exists between their methodologies and ours. Vicuna [1] relies on user-uploaded ShareGPT data for GPT instruction fine-tuning, whereas Alpaca [2] necessitates predefined human prompts for data generation, meticulously curating categories for diversity. In contrast, our approach omits human prompts or user data, initiating QAT fine-tuning data generation with just a single random initial token. To our knowledge, this is yet unexplored in current literature. Our quantization method utilizes standard MinMax quantization (Eq.3).
> [1] Chiang W L, Li Z, Lin Z, et al. Vicuna. Vicuna: An open-source chatbot impressing gpt-4 with 90%* chatgpt quality.
> [2] Taori R, Gulrajani I, Zhang T, et al. Alpaca: A strong, replicable instruction-following model[J]. Stanford Center for Research on Foundation Models.
>
>
> **Q2**: More ablation studies using a subset of each dataset and comparing that to generated data.
> **A2**: Thanks for the suggestion. While the full-training data of the LLaMA model remains unavailable publicly, we utilized the Red Pajama dataset (a substantial LLM pre-training dataset containing 1 trillion tokens) to address concerns. We sampled 100k sentences from this dataset for a fair comparison to our method. In the following table, we obtained an average score of 61.60 on zero-shot reasoning tasks, slightly surpassing fine-tuning on the C4 dataset but still falling short of the results achieved through finetuning on the generated dataset. This result underscores that sampling a small subset from an extensive pretraining dataset struggles to replicate the dataset's distribution adequately. Conversely, the diversity inherent in data extracted from pretrained LLMs via diverse initial tokens proves more suitable for QAT model finetuning. Nonetheless, this comparison holds value and will be included in our manuscript.
>
> | data | #bits | boolq | piqa | siqa | hellaswag | winogrande | arc_easy | arc_challenge | obqa | avg. |
> |-|-|-|-|-|-|-|-|-|-|-|
> | C4 | 4-6-16 |  61.7 | 77.7 | 48.8 | 73.2 | 67.2 | 67.8 | 43.6 | 52.2 | 61.5 |
> | Red Pajama subset | 4-6-16 | 70.0 | 74.9 | 47.1 | 71.2 | 66.1 | 66.8 | 43.4 | 53.0 | 61.6 |
> | generated da​​ta | 4-6-16 | 72.9 | 76.8 | 47.9 | 72.4 | 68.3 | 68.8 | 44.2 | 53.2 | 63.1 |
>
> **Q3**: Have the authors considered using a larger teacher model for the distillation data generation?
> **A3**: Regarding leveraging logits from a superior pre-trained model, we conducted experiments by distilling a LLaMA-7B model using the LLaMA-13B model, shown in the following table. Interestingly, employing the 13B model as the teacher resulted in decreased accuracy compared to using the 7B model. We consider it crucial to employ the same architecture for both teacher and student models. This shared architecture ensures initial weight alignment and promotes the closest distribution match. As a result, the teacher model can effectively guide the student model, aiding in error compensation introduced by quantization.
>
> | #bits | boolq | piqa | siqa | hellaswag | winogrande | arc_easy | arc_challenge | obqa | avg. |  **prev. avg. ** |
> |-|-|-|-|-|-|-|-|-|-|-|
> | 4-8-4 | 69.3 | 74.7 | 46.4 | 67.6 | 63.1 | 63.9 | 42.5 | 49.2 | 59.6 | **60.7** |
> | 4-8-8 | 73.5 | 77.5 | 47.7 | 71.6 | 67.5 | 69.0 | 46.3 | 52.8 | 63.2 | **64.2** |
> | 4-6-16 | 71.2 | 77.1 | 46.5 | 70.6 | 65.4 | 68.1 | 44.7 | 50.6 | 61.8 | **63.1** |
> | 4-8-16 | 73.9 | 77.0 | 48.0 | 71.6 | 67.8 | 69.1 | 46.0 | 54.2 | 63.5 | **64.4** |
> | 4-16-16 | 74.6 | 77.3 | 48.0 | 71.9 | 67.6 | 69.4 | 47.8 | 54.6 | 63.9 | **64.4** |
> | 8-8-4 | 71.1 | 76.4 | 47.1 | 70.1 | 64.9 | 68.2 | 44.0 | 49.2 | 61.4 | **61.6** |
> | 8-8-8 | 74.8 | 79.0 | 47.6 | 73.7 | 70.1 | 71.3 | 47.5 | 56.6 | 65.1 | **66.0** |
> | 8-8-16 | 73.9 | 78.9 | 47.8 | 73.8 | 69.3 | 71.1 | 48.2 | 55.0 | 64.8 | **65.7** |

---

### Official Review · Reviewer_KC6W · 2023-11-01

**Soundness:** 2 fair
**Presentation:** 3 good
**Contribution:** 2 fair
**Rating:** 5
**Confidence:** 4

**Summary:**

The authors study QAT approach for LLMs. The unique challenge of applying QAT on LLMs is the data composition. At the pre-training stage, various different kinds of data sources are mixed and it is difficult to all kinds of data on the fine-tuning stage. To avoid this issue, the authors propose data-free self-distillation which consists of two steps - 1) auto-regressive generation of random sequences from the non-quantized model and 2) knowledge distillation using the generated sequences as inputs to teacher and student models. In the knowledge distillation process, the authors find that soft logit based cross entropy loss works the best. Using the proposed method, the authors quantize LLaMA 7B, 13B and 30B models for various quantization bits (4 and 8 bits for weights, 6, 8 and 16 for activations, 4, 8, 16 for KV cache). The proposed LLM-QAT maintain the quality well for 8 bit weights and activations, while showing some quality degradations for 4 bits. The authors add some ablation studies including comparison between different data source and generated sequences, different quantization methods, and distillation targets.

**Strengths:**

- Overall, the paper is well motivated and well organized.
- The proposed knowledge distillation method is well described and easy to follow.
- Data-free knowledge distillation is a novel approach.

**Weaknesses:**

- Even though the quantized models still maintain the quality closely, there exist non-trivial gaps especially when the weights are quantized to 4-bits. Especially, with additional training (QAT), it is expected the quality is very close to the floating point unquantized model. Also, some PTQ methods such as (Li, Qingyuan, et al. "FPTQ: Fine-grained Post-Training Quantization for Large Language Models." arXiv preprint arXiv:2308.15987 (2023).) claims better quality.

**Questions:**

- In page 4, the activations are weights are -> activations and weights are?
- In page 5, whether the quantize models -> quantized models.
- Why 3-5 tokens? What exact number is used?

---

> ### Author Response · Authors · 2023-11-23
>
> **Q1**: Even though the quantized models still maintain the quality closely, there exist non-trivial gaps especially when the weights are quantized to 4-bits. Especially, with additional training (QAT), it is expected the quality is very close to the floating point unquantized model.
> **A1**: Compared to small models, Language Model Models (LLMs) exhibit distinctive characteristics, where outliers play a significant role. The reviewer's assertion that "additional training (QAT) should yield a quality very close to that of the floating point unquantized model" seems grounded in observations from smaller models, such as BERT or BART. Considering the fact that even for 8-bit quantization, LLMs necessitate special approaches like SmoothQuant [1] or outlier suppression [2], where naive Post-Training Quantization (PTQ) typically suffices for smaller models, it is not surprising that quantizing the LLMs to W4A8 remains a challenging task and still has some gap.
> [1] Xiao, Guangxuan, et al. "Smoothquant: Accurate and efficient post-training quantization for large language models." International Conference on Machine Learning. PMLR, 2023.
> [2] Dettmers, Tim, et al. "LLM.int8(): 8-bit matrix multiplication for transformers at scale." arXiv preprint arXiv:2208.07339 (2022).
>
> **Q2**: Also, some PTQ methods such as (Li, Qingyuan, et al. "FPTQ: Fine-grained Post-Training Quantization for Large Language Models." arXiv preprint arXiv:2308.15987 (2023).) claims better quality.
> **A2**: Regarding the FPTQ paper, it represents concurrent work that is not yet to be formally accepted. Per the ICLR reviewer guidelines, this should not be grounds for rejection. Notably, this paper also shows comparisons with our methodology, as FPTQ adopts a higher granularity, this leads to the higher accuracy.
>
> **Q3**: Typos; Why 3-5 tokens? What exact number is used?
> **A3**: We appreciate your feedback on the typos, all of which will be rectified in our manuscript. To provide specifics, our methodology used {3, 4, 5} deterministic tokens in each round for the first token in 32k vocabulary across three rounds, resulting in a total of 3 x 32k = 96k sentences. Additionally, 4k sentences were generated using random numbers within the range of 3-5. These generated sentences composed our 100k-sample finetuning dataset.

---

### Author Response · Authors · 2023-11-23

Dear Reviewers,

We value your time invested in reviewing our paper and share your curiosity regarding the posed questions. We have diligently prepared a thorough response to address your concerns.

We are encouraged to see that reviewers agree that this approach is novel [Reviewer KC6W] and is the first application of QAT to large language models [Reviewer hBhR]; exhibits overall strong quality and clarity in the presentation [Reviewer WoMH]; and conducts extensive experiments considering several datasets to evaluate [Reviewer dtJe]. However, the authors did not find grounded reasons for the borderline rejection.

Before we dive into the formal rebuttal, let's think about the fascinating aspect of our approach. Have you considered it intriguing that this method is able to fine-tune the QAT model exclusively with generated data while maintaining the LLMs' versatile nature? It is a departure from previous methods like Alpaca [1], which use instruction fine-tuning to make LLMs specialized for particular downstream tasks.

Our method also sets itself apart from concurrent works that utilize generated data for augmentation or random data for calibration. The real challenge lies in preserving the general capabilities of LLMs after fine-tuning, especially after distribution changes stemming from quantization.

We are eager for you to explore our detailed responses, and we welcome further discussion. Your insights are crucial to refining our paper.

[1] Taori R, Gulrajani I, Zhang T, et al. Alpaca: A strong, replicable instruction-following model[J]. Stanford Center for Research on Foundation Models.

---

### Meta-Review · Area_Chair_JUTs · 2023-12-06

**Metareview:**

The paper investigates the data-free quantization-aware training on large language models. It proposes to  generate data from LLMs and studied several sampling strategies. Those data is then used to train quantized models. Experimental validation demonstrated the effectiveness of the approach.


Strengths:
The LLM quantization is an important topic. The empirical study demonstrated that we can use self generated data to train quantized model and obtain good quality quantized models. the paper is generally well written and the technique is validated on multiple datasets.

Weakness:
As pointed out by the reviewers, the use of LLM for data generation is not novel. The empirical study is important but lacks novelty. Ablations between using real datasets vs. generated datasets would be very useful. The authors did replies with additional study showing that generated data is better than sampled real data. It would be better to conduct solid ablation to understand when to what degree the real data can be better than generated data or that can never happen and why.

**Justification For Why Not Higher Score:**

It would be good to have more solid ablation comparing real data vs. generated data to understand the effect of data on QAT.

**Justification For Why Not Lower Score:**

N/A

---

### Decision · Program_Chairs · 2024-01-16

Reject